

# Entanglement dynamics after quantum quenches in generic integrable systems

**Vincenzo Alba⋆ and Pasquale Calabrese**

SISSA and INFN, via Bonomea 265, 34136 Trieste, Italy

⋆ valba@sissa.it

## Abstract

The time evolution of the entanglement entropy in non-equilibrium quantum systems provides crucial information about the structure of the time-dependent state. For quantum quench protocols, by combining a quasiparticle picture for the entanglement spreading with the exact knowledge of the stationary state provided by Bethe ansatz, it is possible to obtain an exact and analytic description of the evolution of the entanglement entropy. Here we discuss the application of these ideas to several integrable models. First we show that for non-interacting systems, both bosonic and fermionic, the exact time-dependence of the entanglement entropy can be derived by elementary techniques and without solving the dynamics. We then provide exact results for interacting spin chains that are carefully tested against numerical simulations. Finally, we apply this method to integrable one-dimensional Bose gases (Lieb-Liniger model) both in the attractive and repulsive regimes. We highlight a peculiar behaviour of the entanglement entropy due to the absence of a maximum velocity of excitations.

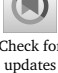

# 1 Introduction

In recent years, understanding the entanglement structure of out-of-equilibrium many-body quantum systems has become an emerging research theme at the crossroad between statistical physics, condensed matter physics, quantum field theory, and quantum information. In one dimension, the growth of entanglement has been related to the capability of a classical computer to simulate non-equilibrium quantum systems with matrix product states (see, e.g., [1–5]). Moreover, the thermodynamic entropy in a stationary state has been interpreted as the asymptotic entanglement of a large subsystem [6–10].

One of the prototype protocols for driving a system out-of-equilibrium is the quantum quench [11–18]: An isolated system is initially prepared at $t = 0$ in a given pure state $|\psi_0\rangle$ (usually the ground state of a quantum many-body hamiltonian $H_0$) and for $t > 0$ the unitary dynamics is governed by a hamiltonian $H$ (with $[H, H_0] \neq 0$ e.g., at $t = 0$ a parameter of the hamiltonian is suddenly changed). Besides the theoretical interest, in recent years it has become possible to investigate quantum quenches experimentally with cold-atom systems [19–30]. Since the post-quench dynamics is unitary, the full system never reaches stationary behaviour, which, instead, can arise locally. The central object to define local equilibration is the reduced density matrix. Given a subsystem $A$ of the full system, the reduced density matrix $\rho_A$ is defined as

$$\rho_A \equiv \text{Tr}_B |\psi\rangle\langle\psi|, \tag{1}$$

where the trace is over the degrees of freedom of the complement $B$ of the subsystem $A$, and $|\psi\rangle \equiv e^{-iHt} |\psi_0\rangle$ is the time-dependent state of the system.

For quantum quenches in generic models, the stationary behaviour of local and quasilocal observables is described by the Gibbs (thermal) ensemble [31–37]. In contrast integrable models possess an extensive number of conserved quantities, besides the hamiltonian, which highly constrain the post-quench dynamics. As a consequence, integrable systems fail to thermalise, meaning that the reduced density matrix for long times is not thermal. Remarkably, a statistical description of local properties of the steady state is possible in terms of a Generalised Gibbs Ensemble (GGE) [12, 16, 17, 38–65], which is obtained by complementing the Gibbs ensemble with all the local and quasilocal conserved quantities [55, 66].

The problem of understanding how entanglement spreads after a quench is deeply intertwined with that of equilibration and thermalisation. The standard measure of the entanglement is the entanglement entropy [67] which is defined as the von Neumann entropy of the reduced density matrix (1):

$$S \equiv -\text{Tr}\rho_A \ln \rho_A. \tag{2}$$

The out-of-equilibrium dynamics of the entanglement entropy following quantum quenches has been the focus of intense research during the last decade [6, 10, 68–89]. Remarkably, in recent years it has become possible to measure entanglement and its evolution in cold-atom experiments [9, 87, 92].

For a wide variety of global quenches, the quasiparticle picture of Ref. [6] provides an understanding of the main qualitative features of the entanglement dynamics. In the quasiparticle picture, the pre-quench initial state is a source of *pairs* of excitations with opposite momentum that travel ballistically through the system. Let us assume that there is only one type of excitations (quasiparticles) identified by their quasi-momentum $\lambda$, and moving with group velocity $v(\lambda)$. The main assumption of the quasiparticle picture is that excitations that are created far apart from each other are incoherent, whereas those emitted at the same point in space are entangled (more precisely, quasi-particles emitted within the initial correlation length, but this refinement just provides a subleading correction to the result [90] and will be ignored in what follows). As the quasiparticles propagate, larger regions of the system get entangled. At time $t$ the entanglement entropy of a subsystem $A$ is proportional to the total number of quasiparticles that after being emitted from the same point in space are shared between subsystem $A$ and its complement. Specifically, for an interval $A$ of length $\ell$ embedded in an infinite one-dimensional system, by counting the quasiparticles with a given weight $s(\lambda)$, one obtains [6]

$$S(t) = 2t \int_{2|v(\lambda)|t<\ell} d\lambda v(\lambda)s(\lambda) + \ell \int_{2|v(\lambda)|t>\ell} d\lambda s(\lambda). \tag{3}$$

Here the function $s(\lambda)$ depends on the production rate of quasiparticles with quasimomentum $\pm\lambda$ and on their individual contribution to the entanglement entropy. Formula (3) holds true in the space-time scaling limit $t, \ell \to \infty$ with the ratio $t/\ell$ fixed. Notice that (3) does not take into account $\mathcal{O}(1)$ terms, which are subleading in the scaling limit. When a maximum quasiparticle velocity $v_M$ exists, such that $|v(\lambda)| \leq v_M$ (e.g., as a consequence of the Lieb-Robinson bound [91]), Eq. (3) predicts that for $t \leq \ell/(2v_M)$, $S$ grows linearly in time because the second term in (3) vanishes. In contrast, for $t \gg \ell/(2v_M)$, only the second term contributes and the entanglement is extensive in the subsystem size, i.e., $S \propto \ell$. Eq. (3) describes the light-cone spreading of the entanglement evolution which has been analytically confirmed in few free models [68–72] and also verified in several numerical studies (see e.g. [76–80]).

However, in order to give some predictive power to (3), we should have a way to fix the entropy density $s(\lambda)$ and the velocity of the entangling quasiparticles $v(\lambda)$. Yet, determining $s(\lambda)$ ab-initio from the dynamical problem is a formidable task even for free models (see e.g. [68]); furthermore, for interacting integrable models, also the identification of the velocity $v(\lambda)$ is a non-trivial issue. A major breakthrough in this respect has been achieved in [10] where it has been shown that, at least for certain classes of quenches in integrable models, the function $s(\lambda)$ can be conjectured from the equivalence between the entanglement and the thermodynamic entropy in the stationary state. The latter can be straightforwardly calculated with equilibrium techniques from the GGE describing the stationary state. In this way Eq. (3) becomes a quantitative analytic conjecture for the entanglement evolution which can be obtained only from the stationary state *without solving the many-body dynamics*. Suggestively, we can state that the main idea of Ref. [10] is to *reconstruct the entanglement evolution going back in time from the stationary state*. Physically, Eq. (3) highlights the transformation during the

dynamics of the entanglement into the thermodynamic entropy. This transformation happens for non-integrable systems as well, but in that case, the entanglement entropy becomes the thermal entropy [7, 9, 93].

In a generic interacting integrable model there are several families of quasiparticles. The generalization of (3) is obtained by summing all the contributions of the different species. The final result of Ref. [10] for the entanglement dynamics is

$$S(t) = \sum_n \Big[ 2t \int\limits_{2|v_n|t<\ell} d\lambda \, v_n(\lambda) s_n(\lambda) + \ell \int\limits_{2|v_n|t>\ell} d\lambda \, s_n(\lambda) \Big], \tag{4}$$

where the index $n$ labels the different families of elementary quasiparticles present in a generic integrable model, and $\lambda$ is their associated momentum label. The sum over the quasiparticle families and momenta reflects the presence in integrable models of well-defined excitations with an infinite lifetime. According to the ideas of Ref. [10] outlined above, in Eq. (4), $s_n(\lambda)$ can be conjectured from the contribution of the individual quasiparticles to the thermodynamic entropy of the GGE describing the steady state. Furthermore, the velocities $v_n(\lambda)$ are assumed to be the group velocities of the low-lying excitations around the steady state. The validity of (4) has been checked numerically for several quenches in the Heisenberg XXZ chain [10]. A generalisation of (4) has been provided to describe the entanglement evolution after *inhomogeneous* quenches in the XXZ chain [94].

In this work we discuss in detail several applications of (4). We start focusing on free fermionic and free bosonic models for which we provide generic results valid for a wide class of quenches. We show that it is possible to recover, in an elementary manner, the known result for the entanglement dynamics after a generic quench in the transverse field Ising chain [68]. For the bosonic case, the quasiparticle picture provides new exact results for the entanglement dynamics in the harmonic chain (the lattice discretisation of the one-dimensional Klein-Gordon field theory). This result is remarkable also because its *ab initio* derivation is not available yet, although we are dealing with a free model. Then, we turn to discuss the entanglement dynamics in the anisotropic Heisenberg chain (XXZ chain). We provide several new theoretical predictions, which complement the results already presented in [10]. For instance, we provide exact results for the post-quench dynamics of the mutual information between two intervals starting from several initial states. This is important because the mutual information is a useful tool to probe the validity of the quasiparticle picture, in which well-defined quasiparticles entangle different regions of the system, so that the mutual information exhibits a peak at intermediate times. An alternative picture is the information scrambling scenario [95–99], which should apply to many non-integrable models such as irrational 1+1 conformal field theories. In the scrambling scenario the quasiparticles loose coherence during the dynamics, due to scattering. As a consequence, for large time the mutual information vanishes independently of the separation of the intervals. Conversely, in integrable models, well-defined quasiparticles exist, and the the mutual information in the space-time scaling limit has a peak also at large times for large enough separation of the intervals, ruling out the scrambling scenario. Numerical evidence supporting the validity of the quasiparticle picture for the mutual information has been provided in [10] considering the quench from the Néel state in the XXZ chain. Moreover, in this work we investigate the signatures of composite excitations (multi-particle bound states) in the mutual information dynamics. An interesting result is that the presence of bound states leads to an anomalous decay of the mutual information at late times and, for some quenches, to multi-peak structures (as already highlighted for other models in [100]).

Another main result obtained here is a quasiparticle prediction for the entanglement dynamics in the one-dimensional Bose gas. We focus on the quench from the Bose-Einstein condensate (BEC), considering both the attractive and repulsive Lieb-Liniger model. In both

cases, at short times the von Neumann entropy exhibits a non-linear increase with time due to the fact there is no maximum velocity of propagation of excitations. Nevertheless, at long times the entanglement entropy saturates. An important difference between attractive and repulsive interactions, is that while for repulsive interactions only one species of quasiparticles is present, for attractive ones multi-boson bound states appear. Interestingly, for weak interactions, bound states contribute significantly to the entanglement dynamics. Moreover, similar to the XXZ chain, their presence is reflected in a slow vanishing behaviour of the mutual information between two intervals at late times.

The outline of the paper is as follows. Section 2 is devoted to the entanglement dynamics after quantum quenches in free-fermion and free-boson models. In section 3, we detail the approach of [10] for the entanglement dynamics in a generic Bethe ansatz integrable model. In section 4 we provide several results for the entanglement dynamics in the XXZ chain. In section 5 we present the quasiparticle results for the entanglement dynamics after the quench from the Bose-Einstein condensate in the Lieb-Liniger model. In the last section we discuss several points and developments which deserve further investigation.

## 2 Entanglement dynamics in free models

In this section we employ the quasiparticle scenario of [10] to derive analytically the entanglement dynamics in free-fermion and free-boson models after rather generic quenches. We test these results against exact analytical and numerical results for the entanglement dynamics after a global quench in the transverse field Ising/XY chain and in the harmonic chain. These models can be mapped onto a system of free fermions and free bosons, respectively. For the Ising model our result agrees with the ab initio derivation in [68], providing a further benchmark of the ideas pursued in this paper and in [10]. For the harmonic chain our results have been anticipated in [101] and appeared, for a similar bosonic model, also in [102].

### 2.1 Models of free fermions

If a translational invariant fermionic model is free, it means that the hamiltonian in momentum space can be mapped into (apart from an unimportant additive constant)

$$H = \sum_k \epsilon_k b_k^\dagger b_k, \tag{5}$$

where $b_k$ are fermionic mode occupation operators satisfying standard anticommutation relations and $\epsilon_k$ is the energy of the mode $k$ (i.e. the dispersion relation).

For all these models, the GGE built with local conservation laws is equivalent to the one built with the mode occupation numbers $\hat{n}_k = b_k^\dagger b_k$ since they are linearly related [45, 49]. Thus the local properties of the stationary state are captured by the GGE density matrix

$$\rho_{\mathrm{GGE}} \equiv \frac{e^{-\sum_k \lambda_k \hat{n}_k}}{Z}, \tag{6}$$

where $Z = \mathrm{Tr} e^{-\sum_k \lambda_k \hat{n}_k}$ ensures the normalisation $\mathrm{Tr} \rho_{\mathrm{GGE}} = 1$.

The thermodynamic entropy of the GGE is obtained by elementary methods, leading, in the thermodynamic limit, to

$$S_{\mathrm{TD}} = L \int \frac{dk}{2\pi} H(n_k), \tag{7}$$

where $n_k \equiv \langle \hat{n}_k \rangle_{\mathrm{GGE}} = \mathrm{Tr}(\rho_{\mathrm{GGE}} \hat{n}_k)$ and the function $H$ is

$$H(n) = -n \ln n - (1-n) \ln(1-n). \tag{8}$$

The interpretation of Eq. (7) is obvious: the mode $k$ is occupied with probability $n_k$ and empty with probability $1 - n_k$. Given that $\hat{n}_k$ is an integral of motion, one does not need to compute explicitly the GGE (6), but it is sufficient to calculate the expectation values of $\hat{n}_k$ in the initial state $\langle \psi_0 | \hat{n}_k | \psi_0 \rangle$ which, by construction, equals $n_k = \langle \hat{n}_k \rangle_{\text{GGE}}$.

At this point, following [10], we identify the stationary thermodynamic entropy with the density of entanglement entropy to be plugged in Eq. (3), obtaining the general prediction

$$S(t) = 2t \int\limits_{2|\epsilon'_k|t<\ell} \frac{dk}{2\pi} \epsilon'_k H(n_k) + \ell \int\limits_{2|\epsilon'_k|t>\ell} \frac{dk}{2\pi} H(n_k), \tag{9}$$

where $\epsilon'_k = d\epsilon_k/dk$ is the group velocity of the mode $k$. This formula is generically valid for arbitrary models of free fermions with the *crucial but rather general* assumption that the initial state is writable in terms of *pairs* of quasiparticles. More general and peculiar structures of initial states can be also considered, see [103, 104].

### 2.1.1 Test for the transverse field Ising chain

Eq. (9) can be tested against available exact analytic results for the transverse field Ising chain with hamiltonian

$$H = -\sum_{j=1}^{L} [\sigma_j^x \sigma_{j+1}^x + h\sigma_j^z], \tag{10}$$

where $\sigma_j^{x,z}$ are Pauli matrices and $h$ is the transverse magnetic field. We use periodic boundary conditions in (10).

The hamiltonian (10) is diagonalised by a combination of Jordan-Wigner and Bogoliubov transformations, leading to Eq. (5) where the single-particle energies are

$$\epsilon_p = 2\sqrt{1 + h^2 - 2h\cos p}. \tag{11}$$

We focus on a quench of the magnetic field in which the chain is initially prepared in the ground state of (10) with $h_0$ and then, at $t = 0$ the magnetic field is suddenly changed from $h_0$ to $h$. As in the general analysis above, the steady-state is determined by the fermionic occupation numbers $n_k$ given by [45, 105]

$$n_k = \frac{1}{2}(1 - \cos\Delta_k), \tag{12}$$

where $\Delta_k$ is the difference of the pre- and post-quench Bogoliubov angles [105]

$$\Delta_p = \frac{4(1 + hh_0 - (h + h_0)\cos p)}{\epsilon(p)\epsilon_0(p)}, \tag{13}$$

where $\epsilon_0(p)$ and $\epsilon(p)$ stand for pre- and post-quench dispersion relations respectively.

The quasiparticle prediction for the entanglement dynamics after the quench is then Eq. (9) with $n_k$ in (12). This coincides with the *ab initio* derivation performed in [68]. The same derivation is valid also for a generic quench in the XY chain reported in [68].

### 2.2 Free bosonic models

For a free bosonic model, the hamiltonian can be written after some suitable transformations as (apart from a unimportant additive constant)

$$H = \sum_k \epsilon_k a_k^\dagger a_k. \tag{14}$$

with $[a_k, a_{k'}^\dagger] = \delta_{k,k}$ being bosonic mode operators.

The stationary values of local observables can be described by a generalised Gibbs ensemble (GGE) constructed from the mode occupation numbers $\hat{n}_k = a_k^\dagger a_k$ with the GGE density matrix

$$\rho_{\text{GGE}} = Z^{-1} e^{-\sum_k \lambda_k \hat{n}_k}, \tag{15}$$

where $\lambda_k$ are Lagrange multipliers and $Z$ is a normalisation. The Lagrange multipliers $\lambda_k$ in (15) are fixed by imposing that the expectation value of $\hat{n}_k$ in the initial state coincides with its GGE average. The initial value $n_k \equiv \langle \psi_0 | \hat{n}_k | \psi_0 \rangle$ is easily calculated from the initial state. The GGE expectation value of $\hat{n}_k$ is obtained as

$$\langle \hat{n}_k \rangle_{\text{GGE}} = \text{Tr}[\hat{n}_k \rho_{\text{GGE}}] = -\frac{\partial}{\partial \lambda_k} \ln Z, \tag{16}$$

with

$$Z = \text{Tr} e^{-\sum_k \lambda_k \hat{n}_k} = \prod_k \sum_{n_k=0}^\infty e^{-\lambda_k n_k} = \prod_k \frac{1}{1 - e^{-\lambda_k}}. \tag{17}$$

Thus, one has

$$\langle \hat{n}_k \rangle_{\text{GGE}} = \frac{\partial}{\partial \lambda_k} \sum_k \ln(1 - e^{-\lambda_k}) = \frac{1}{e^{\lambda_k} - 1}. \tag{18}$$

After imposing the conservation of $\hat{n}_k$, i.e., that (18) equals $n_k = \langle \psi_0 | \hat{n}_k | \psi_0 \rangle$, one obtains $\lambda_k$ as

$$e^{\lambda_k} = 1 + n_k^{-1}. \tag{19}$$

At this point, calculating the thermodynamic entropy is a trivial exercise in statistical physics:

$$S_{\text{GGE}} = -\text{Tr} \rho_{\text{GGE}} \ln \rho_{\text{GGE}} = -\text{Tr} \frac{e^{-\sum_k \lambda_k \hat{n}_k}}{Z} \ln \frac{e^{-\sum_k \lambda_k \hat{n}_k}}{Z} \tag{20}$$

$$= \text{Tr}\Big[\rho_{\text{GGE}} \Big(\sum_k \lambda_k \hat{n}_k + \ln Z\Big)\Big] = \sum_k -\lambda_k \frac{\partial \ln Z}{\partial \lambda_k} + \ln Z. \tag{21}$$

Using that $Z = \prod_k (1 - e^{-\lambda_k})^{-1}$ (cf. Eq. (17)), we obtain

$$S_{\text{GGE}} = \sum_k \frac{\lambda_k}{e_k^\lambda - 1} + \ln(1 - e^{-\lambda_k}) = \sum_k (n_k + 1)\ln(n_k + 1) - n_k \ln n_k, \tag{22}$$

where we used $n_k = 1/(e^{\lambda_k} - 1)$, cf. (19). In the thermodynamic limit the sum over the momenta becomes an integral and (22) becomes

$$S_{GGE} = L \int_{-\pi}^\pi \frac{dk}{2\pi}[(n_k + 1)\ln(n_k + 1) - n_k \ln n_k] \equiv L \int_{-\pi}^\pi dk\, s(k), \tag{23}$$

where in the rightmost side of the equation we introduced the entropy contribution $s(k)$ of the quasiparticle with momentum $k$ as

$$2\pi s(k) = (n_k + 1)\ln(n_k + 1) - n_k \ln n_k. \tag{24}$$

At this point, we are ready to use the fact that the entanglement entropy is the stationary thermodynamic entropy (23) so that the quasiparticle picture for the entanglement evolution (4) gives

$$S_A(t) = t \int_{2|v_k|t < \ell} dk\, s(k) 2|v_k| + \ell \int_{2|v_k|t > \ell} dk\, s(k), \tag{25}$$

where the entropy density $s(k)$ is given by (24) and $v_k = d\epsilon_k/dk$.

### 2.2.1 Tests for the harmonic chain

Here we focus on one of the simplest bosonic models with an exactly solvable non-equilibrium dynamics, i.e., the harmonic chain defined by the hamiltonian

$$H = \frac{1}{2} \sum_{n=0}^{N-1} \left[ \pi_n^2 + m^2 \phi_n^2 + (\phi_{n+1} - \phi_n)^2 \right], \tag{26}$$

with periodic boundary conditions. Eq. (26) defines a chain of $N$ harmonic oscillators with frequency (mass) $m$ and with nearest-neighbour quadratic interactions. Here $\phi_n$ and $\pi_n$ are the position and the momentum operators of the $n$-th oscillator, with equal time commutation relations

$$[\pi_m, \pi_n] = i\delta_{nm}, \qquad [\phi_n, \phi_m] = [\pi_n, \pi_m] = 0. \tag{27}$$

In the context of quench dynamics the harmonic chain was first discussed in [11] to which we refer for a detailed analysis; here we only report the results relevant for our aims. The harmonic chain is easily diagonalised in momentum space where it assumes the standard diagonal form (14) with dispersion relation

$$\epsilon_k^2 = m^2 + 2(1 - \cos k). \tag{28}$$

We now consider the quantum quench in which the harmonic chain is initially prepared in the ground-state $|\psi_0\rangle$ of (26) with $m = m_0$, and at time $t = 0$ the mass is quenched to a different value $m \neq m_0$. We use the notation $\epsilon_k^0$ for the dispersion relation in the initial state and $\epsilon_k$ for the one for $t > 0$.

In order to give predictive power to Eq. (25) we just need to fix the conserved value $n_k = \langle \psi_0 | \hat{n}_k | \psi_0 \rangle$ which is obtained by elementary methods [12]

$$n_k = \langle \psi_0 | \hat{n}_k | \psi_0 \rangle = \langle \psi_0 | a_k^\dagger a_k | \psi_0 \rangle = \frac{1}{4} \left( \frac{\epsilon_k}{\epsilon_k^0} + \frac{\epsilon_k^0}{\epsilon_k} \right) - \frac{1}{2}. \tag{29}$$

Also the group velocity from (28) is

$$v_k = \frac{d\epsilon_k}{dk} = \frac{\sin k}{\sqrt{m^2 + 2(1 - \cos(k))}}. \tag{30}$$

The quasiparticle prediction (cf. (25)) for the entanglement dynamics after the mass quench in the harmonic chain is reported in Figure 1. The Figure shows the entropy density $S(t)/\ell$ plotted versus the rescaled time $t/\ell$, with $\ell$ the size of subsystem $A$. The different curves in the Figure correspond to quenches with different values of $m$, namely $m = 2$ (continuous line), $m = 3$ (dashed-dotted line), $m = 5$ (dotted line). The pre-quench value of the mass is fixed to $m_0 = 1$. The results are obtained using (25).

The entanglement entropy exhibits the expected linear behaviour at short times followed by a saturation at asymptotically long times. Clearly, the steady-state value of the entanglement entropy increases with $m$. In the limit $m \gg m_0$, the steady-state entropy at the leading order in $1/m$ is $S \approx \ln m$. The crossover time from the linear to the saturation regime increases with $m$, because the maximum velocity $v_M$ decreases upon increasing $m$, as it is clear from (30).

### 2.2.2 Numerical checks

We now provide numerical checks of the validity of (25). The entanglement dynamics after a global quench in the harmonic chain has been studied numerically in several papers [70,

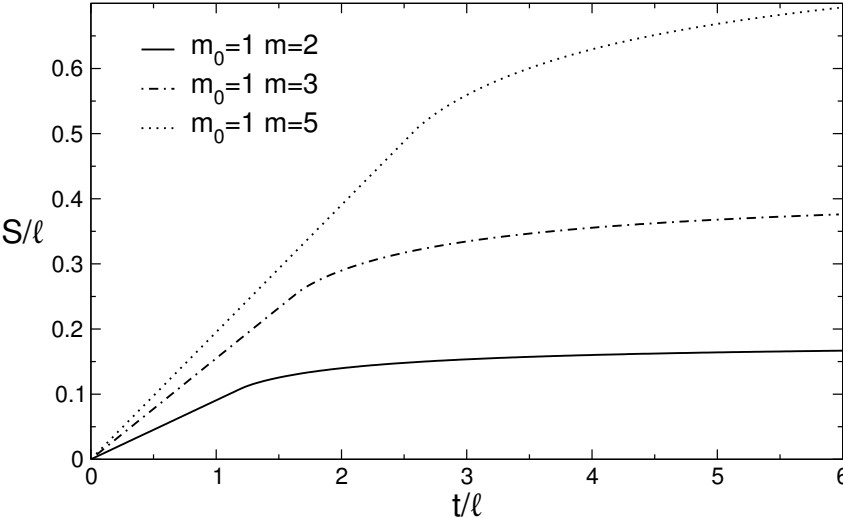

Figure 1: Entanglement dynamics after a mass quench in the harmonic chain: Theoretical prediction using the quasiparticle picture. The entropy density $S/\ell$ is plotted against the rescaled time $t/\ell$, with $\ell$ the size of $A$. Different lines are results for quenches with different values of the chain mass $m$. The pre-quench value of the mass $m_0 = 1$ is the same for all the quenches.

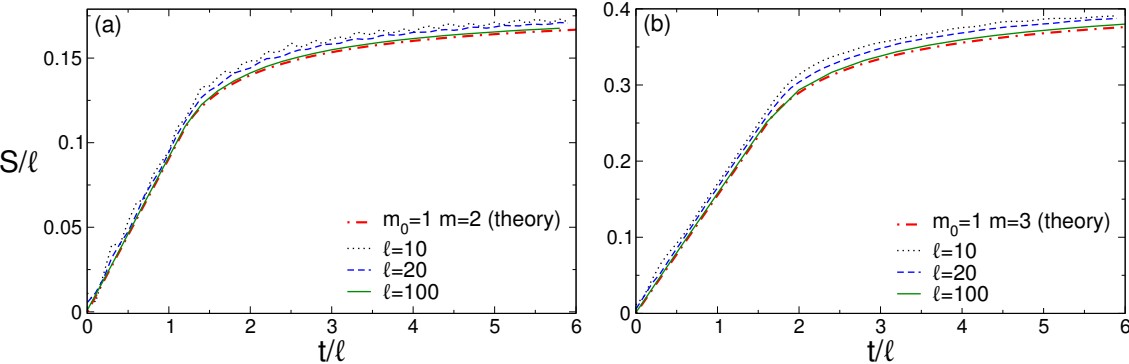

Figure 2: Entanglement dynamics after a mass quench in the harmonic chain: Comparison between the quasiparticle picture and finite-chain results. In both panels the entropy density $S/\ell$ is plotted against the rescaled time $t/\ell$, with $\ell$ the size of $A$. Panels (a) and (b) show results for the quenches with final mass $m = 2$ and $m = 3$, respectively. The pre-quench value of the mass is $m_0 = 1$. In both panels dotted, dashed, and continuous lines are finite-size results for a chain with $L = 1000$ sites and subsystem sizes $\ell = 10, 20, 100$. The dashed-dotted line is the prediction obtained using the quasiparticle picture in the space-time scaling limit.

83, 84]. These papers focused on the critical ($m \to 0$) and continuum limit, in which several simplifications occur because there is a single velocity of excitations. The quasi-particle prediction turned out to be correct, but with additive logarithmic corrections due to the presence of a zero mode [84]. In the following we focus on the massive regime that so far received only little attention.

For systems of free bosons, at any time after the quench the entanglement entropy of a finite subsystem can be calculated effectively [106, 107] from the time-dependent two-point correlation functions reported in [12]. In Figure 2 we present numerical results for the entanglement entropy $S(t)$ after a mass quench in the harmonic chain. The results are for a chain with $L = 1000$ sites and subsystems sizes $\ell = 10, 20, 100$. We numerically checked that for these values of $\ell$ the effect of the finite $L$ is negligible. The two panels (a) and (b) show

results for the quenches with $m = 2$ and $m = 3$, respectively. The pre-quench value of the mass $m_0 = 1$ is the same for both quenches. The theoretical prediction obtained using the quasiparticle picture (cf. (25)) is reported in the Figure as dashed-dotted line. For any finite $\ell$ scaling corrections are expected because Eq. (25) holds only in the space-time scaling limit with $\ell, t \to \infty$, at $t/\ell$ fixed. These corrections are clearly visible in the data. However, they rapidly decrease upon increasing $\ell$, and the results for $\ell = 100$ are almost indistinguishable from the thermodynamic limit predictions.

# 3 Entanglement dynamics in a generic Bethe ansatz integrable model

In this section, following the ideas of [10], we show how the quasiparticle prediction (4) can be applied to a generic Bethe ansatz integrable model. In order to do so, in the next two subsections we provide explicit conjectures for the values of $s_n(\lambda)$ and $v_n(\lambda)$ to be plugged in (4). As explained in the introduction, $s_n(\lambda)$ can be read off from the thermodynamic entropy in the stationary state that can be worked out in the thermodynamic Bethe ansatz framework. For $v_n(\lambda)$, we will instead use the velocity of low-lying particle-hole excitations built on top of the stationary state. In the following subsections, we will show how to derive these velocities by Bethe ansatz techniques following Ref. [108].

## 3.1 The thermodynamic Bethe ansatz

In a Bethe anstaz integrable model of length $L$, with $N$ elementary particles, and with periodic boundary conditions, the eigenstates are in one to one correspondence with a set of $N$ complex quasi-momenta $\lambda_j$ (known as rapidities) which satisfy model dependent quantisation conditions denoted as Bethe equations (here we focus on models with an "elementary" Bethe ansatz; there are models with more than one type of rapidities leading to the so-called nested Bethe ansatz [109]; in that case the modification of (4) is straightforward because one has just to perform a further sum on the types of the rapidities, see [100] for an illustrative example). The prototype integrable model that we consider here is the XXZ spin-1/2 chain in the regime with $\Delta > 1$, although the TBA results that we will discuss can be generalized to the case with $\Delta < 1$ and to other integrable models with minor modifications. In the thermodynamic limit and for a generic translational invariant model, the vast majority of the solutions of the Bethe equations obey the string hypothesis [110]. Specifically, solutions of the Bethe equations form string patterns in the complex plane. Rapidities forming a $n$-string are parametrised as [110]

$$\lambda_{n,\gamma}^j = \lambda_{n,\gamma} + i\frac{\eta}{2}(n + 1 - 2j) + \delta_{n,\gamma}^j, \tag{31}$$

where $\eta$ is an interaction parameter, $j = 1, \ldots, n$ labels the different string components, $\lambda_{n,\gamma}$ is the "string centre", and $\delta_{n,\gamma}^j$ are the string deviations, which for the majority of the eigenstates are $\delta_{n,\gamma}^j = \mathcal{O}(e^{-L})$, implying that they can be neglected in the thermodynamic limit (string hypothesis [110]). Physically, a $n$-string corresponds to a bound state of $n$ elementary particles. For the XXZ chain with $\Delta < 1$ the structure of the string solutions is more complicated [110] than (31), although major simplifications occur for $\Delta = \Delta_k \equiv -\cos(\pi/k)$ with $k = 1, 2, \ldots$ (roots of unity).

Within the framework of the string hypothesis, the string centres $\lambda_{n,\gamma}$ are obtained by solving the Bethe-Gaudin-Takahashi (BGT) equations [110]

$$L\pi_n(\lambda_{n,\alpha}) = 2\pi I_{n,\alpha} + \sum_{(n,\alpha)\neq(m,\beta)} \Theta_{n,m}(\lambda_{n,\alpha} - \lambda_{m,\beta}). \tag{32}$$

Here $I_{n,\alpha}$ are (integer or half-integer) quantum numbers, $\pi_n(x)$ are model dependent functions for the string momentum. The scattering phases for the bound states $\Theta_{n,m}(\lambda)$ can be written as

$$\Theta_{n,m}(\lambda) \equiv (1-\delta_{n,m})\theta_{|n-m|}(\lambda) + 2\theta_{|n-m|+2}(\lambda) + \cdots + \theta_{n+m-2}(\lambda) + \theta_{n+m}(\lambda), \quad (33)$$

in terms of a model dependent elementary phase shift $\theta_n(\lambda)$. Each different choice of $I_{n,\alpha}$ identifies a different set of solutions of (32), which correspond to a different eigenstate of the considered integrable model. The corresponding eigenstate energy $E$ and total momentum $P$ are obtained by summing over all the BGT rapidities [110] as

$$E = \sum_{n,\alpha} \epsilon_n(\lambda_{n,\alpha}), \qquad P = \sum_{n,\alpha} \pi_n(\lambda_{n,\alpha}) = \sum_{n,\alpha} z_n(\lambda_{n,\alpha}), \quad (34)$$

where $\epsilon_n(\lambda)$ is the model dependent string energy, while $z_n(\lambda_{n,\alpha}) = 2\pi I_{n,\alpha}/L$ so that the total momentum $P$ depends only on the $I_{n,\alpha}$.

In the thermodynamic limit the solutions of the BGT equations (32) become dense on the real axis. The central quantities to describe local properties of the system are then the rapidity densities $\rho_n(\lambda)$ ($n$ labelling different string types) which are formally defined in the thermodynamic limit as

$$\rho_n(\lambda) \equiv \lim_{L \to \infty} \frac{1}{L(\lambda_{n,\gamma+1} - \lambda_{n,\gamma})}. \quad (35)$$

To fully specify the thermodynamic state of the system, the densities $\rho_n^{(h)}(\lambda)$ of the $n$-string holes, i.e., of the unoccupied string centres are also required. Finally, it is also custom [110] to introduce the total densities $\rho_n^{(t)}(\lambda) \equiv \rho_n(\lambda) + \rho_n^{(h)}(\lambda)$. Some TBA relations are written in a more compact form in terms of the ratio

$$\eta_n(\lambda) \equiv \frac{\rho_n^{(h)}(\lambda)}{\rho_n(\lambda)}, \quad (36)$$

that we introduce for future convenience.

The $\rho_n^{(h)}(\lambda)$ and $\rho_n(\lambda)$ are obtained via the thermodynamic version of the BGT equations

$$\rho_n^{(h)}(\lambda) + \rho_n(\lambda) = b_n(\lambda) - \sum_{m=1}^{\infty} (a_{nm} \star \rho_m)(\lambda), \quad (37)$$

which are obtained from (32) by taking the thermodynamic limit. The symbol $f \star g$ denotes the convolution between two functions as

$$(f \star g)(\lambda) = \int_{-\pi/2}^{\pi/2} d\mu f(\lambda - \mu) g(\mu). \quad (38)$$

The functions $b_n(\lambda)$ and $a_{nm}(\lambda)$ are related to $\pi_n(\lambda)$ and $\Theta_{nm}(\lambda)$ as

$$b_n(\lambda) \equiv \frac{1}{2\pi} \frac{d\pi_n(\lambda)}{d\lambda}, \qquad a_{nm}(\lambda) \equiv \frac{1}{2\pi} \frac{d\Theta_{nm}(\lambda)}{d\lambda}. \quad (39)$$

In the thermodynamic limit, the expectation values of local conserved quantities are functionals of the densities $\rho_n(\lambda)$; for example the particle and energy densities are

$$\frac{N}{L} = \sum_{n=1}^{\infty} n \int d\lambda \rho_n(\lambda), \quad (40)$$

$$\frac{E}{L} = \sum_{n=1}^{\infty} \int d\lambda \epsilon_n(\lambda) \rho_n(\lambda). \quad (41)$$

The set of rapidity densities $\boldsymbol{\rho} \equiv \{\rho_n\}_{n=1}^{\infty}$ defines a thermodynamic macrostate, which encodes all the expectation values of local or quasi-local observables in the thermodynamic limit. A generic thermodynamic macrostate corresponds to an exponentially large (with $L$) number of microscopic eigenstates of the model, all leading to the same set of rapidity densities in the thermodynamic limit. The total number of possible choices is $e^{S_{YY}}$, with $S_{YY}$ the Yang-Yang entropy [111]

$$S_{YY}[\boldsymbol{\rho}] \equiv L \sum_{n=1}^{\infty} \int d\lambda \Big[ \rho_n^{(t)} \ln \rho_n^{(t)} - \rho_n \ln \rho_n - \rho_n^{(h)} \ln \rho_n^{(h)} \Big]. \tag{42}$$

The Yang-Yang entropy represents the thermodynamic entropy of a given macrostate, as it should be clear from a generalised microcanonical argument. For example, it has been proved that for systems in thermal equilibrium $S_{YY}$ coincides with the thermal entropy [110]. Our conjecture for the time evolution of the entanglement starts from the Yang-Yang entropy since we assume that at long times the entanglement entropy is the thermodynamic one. Furthermore, we also assume that the Bethe quasiparticles are the one entangling the system and appearing in (4). Thus it is natural to identify $s_n(\lambda)$ with the integrand in (42), i.e.

$$s_n(\lambda) = \rho_n^{(t)} \ln \rho_n^{(t)} - \rho_n \ln \rho_n - \rho_n^{(h)} \ln \rho_n^{(h)}. \tag{43}$$

Here the three sets of root densities $\rho_n$, $\rho_n^{(h)}$, and $\rho_n^{(t)}$ refer to the macrostate that describes the stationary state. This is in principle calculable by Bethe ansatz techniques from the overlaps of the initial state with the Bethe states [112, 113] or equivalently from the GGE [55].

## 3.2 Group velocities over a macrostate

Having identified $s_n(\lambda)$ in Eq. (4), the other crucial ingredient for the quasiparticle picture for the entanglement dynamics is the group velocity of the entangling quasiparticles. In the approach of [10] the entangling quasiparticles are identified with the low-lying excitations (particle-hole excitations) around the thermodynamic macrostate describing the steady state.

The low-lying excitations over a given macrostate can be constructed explicitly in the framework of TBA as originally pointed out for the stationary state after a quench in [108] and only briefly summarised in the following. The first step is to choose, among the equivalent eigenstates of the macrostate identified by the densities $\rho_n, \rho_n^{(h)}$, one representative microstate at finite, but large, volume $L$. This corresponds to a particular set of BGT quantum numbers $I_{n,\alpha}$ in (32) chosen in such a way that the resulting rapidities from the BGT equations are a discretisation of the desired macrostate. A particle-hole excitation in each $n$-string sector is obtained by replacing $I_{n,h} \to I_{n,p}$, where $I_{n,p}(I_{n,h})$ is the BGT number of the new added particle (hole). Due to interactions, this local change in quantum numbers implies a rearrangement of *all* the rapidities. The excess energy of the particle-hole excitation is easily calculated as

$$\delta E_n = e_n(\lambda_{n,p}) - e_n(\lambda_{n,h}). \tag{44}$$

Remarkably, apart from the dressing of the "single-particle" energy $e(\lambda)$ (44) is the same as for free models. Similarly, the change in the total momentum is obtained from (54) as

$$\delta K_n = z_n(\lambda_{n,p}) - z_n(\lambda_{n,h}). \tag{45}$$

Finally, the group velocity of the particle-hole excitations is by definition

$$v_n(\lambda) \equiv \frac{\delta E_n}{\delta K_n} = \frac{\partial e_n}{\partial z_n} = \frac{e_n'(\lambda)}{z_n'(\lambda)} = \frac{e_n'(\lambda)}{2\pi \rho_n (1 + \eta_n(\lambda))}. \tag{46}$$

Here we used that $dz_n(\lambda)/d\lambda = 2\pi\rho_n^{(t)}$, with $\rho_n^{(t)} \equiv \rho_n(1 + \eta_n)$. The function $e'_n(\lambda)$ is determined by solving an infinite system of Fredholm integral equations of the second kind as

$$e'_n(\lambda) + \frac{1}{2\pi} \sum_{m=1}^{\infty} \int d\mu\, e'_m(\mu) \frac{\Theta'_{m,n}(\mu-\lambda)}{1+\eta_m(\mu)} = \epsilon'_n(\lambda). \tag{47}$$

Equations (47) are routinely solved numerically by truncating the system, i.e., considering $n \le n_{max}$ and checking convergence with varying $n_{max}$. The method outlined above for calculating the group velocities has been introduced in [108] in order to study velocity of the spreading of correlation after a quench from a thermal state. Very recently it has also been used to study transport properties in integrable models [114–117].

At this point, we have a Bethe ansatz procedure to calculate the velocities of the entangling quasiparticles and we are ready to use the conjecture (4) to provide quantitative predictions for the entanglement spreading in generic integrable systems.

## 4 Entanglement dynamics in Heisenberg spin chains

In this section we focus on the spin-1/2 anisotropic Heisenberg chain (XXZ chain). The goal of this section is to provide a thorough discussion of some results that have been already presented in [10] and to extend them in several directions.

The XXZ chain is defined by the hamiltonian

$$\mathcal{H} = \sum_{i=1}^{L} \Big[ \frac{1}{2}(S_i^+ S_{i+1}^- + S_i^+ S_{i+1}^-) + \Delta\Big(S_i^z S_{i+1}^z - \frac{1}{4}\Big)\Big]. \tag{48}$$

Here $S_i^\alpha$ are spin-1/2 operators acting at site $i$ of the chain, and $\Delta$ is the anisotropy parameter. Periodic boundary conditions are used in (48).

We focus on the non-equilibrium dynamics ensuing from several low-entangled initial states, namely the tilted Néel state

$$|\vartheta, \nearrow\swarrow \cdots\rangle \equiv e^{i\vartheta \sum_j S_j^y} |\uparrow\downarrow \cdots\rangle, \tag{49}$$

the Majumdar-Ghosh (dimer) state

$$|MG\rangle \equiv ((|\uparrow\downarrow\rangle - |\downarrow\uparrow\rangle)/2)^{\otimes L/2}, \tag{50}$$

and the tilted ferromagnet

$$|\vartheta, \nearrow\nearrow\rangle \equiv e^{i\vartheta \sum_j S_j^y} |\uparrow\uparrow \cdots\rangle, \tag{51}$$

Here $\vartheta$ is the tilting angle.

The results that we obtain here build on a large literature about the integrable quench dynamics of the XXZ chain [52–55, 118–129] to which we refer for completeness.

### 4.1 Bethe ansatz solution of the $XXZ$ chain

In the Bethe ansatz solution of the $XXZ$ chain, the eigenstates of (48) can be labeled by the total number of down spins (particles). Eigenstates in the sector with $M$ particles are in correspondence with a set of $M$ rapidities $\lambda_j$. The rapidities are obtained by solving a set of non linear algebraic equations (Bethe equations) as [110]

$$\Big[ \frac{\sin(\lambda_j + i\frac{\eta}{2})}{\sin(\lambda_j - i\frac{\eta}{2})} \Big]^L = -\prod_{k=1}^{M} \frac{\sin(\lambda_j - \lambda_k + i\eta)}{\sin(\lambda_j - \lambda_k - i\eta)}, \tag{52}$$

where $\eta \equiv \mathrm{arccosh}(\Delta)$. In the thermodynamic limit the vast majority of the solutions of the Bethe equations (52) organise according to the string hypothesis (31). For the XXZ spin-chain, physically, a $n$-string corresponds to a bound states of $n$ down spins. The BGT equations are given in (32) in which one should identify

$$\theta_n(\lambda) = \pi_n(\lambda) = 2\arctan\Big[\frac{\tan(\lambda)}{\tanh(n\eta/2)}\Big]. \tag{53}$$

For $\Delta > 1$, the string centres are in the interval $[-\pi/2, \pi/2)$. The eigenstate energy $E$ and total momentum $P$ are given by Eq. (34) with string energy

$$\epsilon_n(\lambda) \equiv -\frac{\sinh(\eta)\sinh(n\eta)}{\cosh(n\eta) - \cos(2\lambda)}, \tag{54}$$

The thermodynamic version of the BGT equations are given by (37).

We also consider the XXZ chain in the limit $\Delta = 1$ (XXX chain). The Bethe ansatz results for the XXX chain can be obtained from those for the XXZ chain by taking an appropriate scaling limit. The first step is to rewrite the formulas for the XXZ chain in terms of the rescaled rapidities $\mu$ defined as

$$\mu \equiv \frac{\lambda}{\eta}. \tag{55}$$

As $\eta \to 0$, i.e., for $\Delta \to 1$, the rescaled rapidities $\mu$ are now defined in the whole real axis $[-\infty, \infty]$. Moreover, the spacing along the imaginary axis between different string components is $i/2$. Using (54) one obtains that for the XXX chain $\epsilon_n(\mu)$ is

$$\epsilon_n(\mu) = \frac{2n}{4\mu^2 + n^2}. \tag{56}$$

## 4.2 Thermodynamic Bethe ansatz for global quenches

In the TBA approach for quantum quenches, local and quasilocal properties of the post-quench steady state are described by an appropriate thermodynamic macrostate [112, 113]. This macrostate is fully characterised by its rapidity densities $\rho_n(\lambda)$ and $\rho_n^{(h)}(\lambda)$ (or equivalently $\eta_n(\lambda)$). For all initial states considered here (cf. (49) (50) (51)) the macrostate densities satisfy the recursive relations

$$\eta_n(\lambda) = \frac{\eta_{n-1}(\lambda - i\frac{\eta}{2})\eta_{n+1}(\lambda + i\frac{\eta}{2})}{1 + \eta_{n-2}(\lambda)} - 1, \tag{57}$$

$$\rho_n^{(h)}(\lambda) = \rho_{n-1}^{(t)}(\lambda + i\frac{\eta}{2}) + \rho_{n-1}^{(t)}(\lambda - i\frac{\eta}{2}) - \rho_{n-1}^{(h)}(\lambda), \tag{58}$$

with initial conditions $\eta_0 = 0$ and $\rho_0^{(h)} = 0$. The information on the pre-quench initial state is encoded in the densities $\rho_1^{(h)}$ and $\eta_1$.

For completeness, we report the results for the quenches considered in this work. For the tilted ferromagnet one has [126]

$$\eta_1(\lambda) = -1 + \frac{T_1\left(\lambda + i\frac{\eta}{2}\right)T_1\left(\lambda - i\frac{\eta}{2}\right)}{\phi\left(\lambda + i\frac{\eta}{2}\right)\bar{\phi}\left(\lambda - i\frac{\eta}{2}\right)}, \tag{59}$$

$$\rho_1^{(h)}(\lambda) = \frac{\sinh\eta}{\pi}\Big(\frac{1}{\cosh(\eta) - \cos(2\lambda)}$$

$$-\frac{2\sin^2(\vartheta)\{2\sin^2(\vartheta) + \cosh(\eta)[(\cos(2\vartheta) + 3)\cos(2\lambda) + 4]\}}{\sinh^2(\eta)[\cos(2\vartheta) + 3]^2\sin^2(2\lambda) + \{2\sin^2(\vartheta) + \cosh(\eta)[(\cos(2\vartheta) + 3)\cos(2\lambda) + 4]\}^2}\Big), \tag{60}$$

where

$$T_1(\lambda) = \cos(\lambda)\left(4\cosh(\eta) - 2\cos(2\vartheta)\sin^2\lambda + 3\cos(2\lambda) + 1\right), \tag{61}$$

$$\phi(\lambda) = 2\sin^2\vartheta \sin\lambda \cos\left(\lambda + i\frac{\eta}{2}\right)\sin\left(\lambda - i\frac{\eta}{2}\right), \tag{62}$$

$$\bar{\phi}(\lambda) = 2\sin^2\vartheta \sin\lambda \cos\left(\lambda - i\frac{\eta}{2}\right)\sin\left(\lambda + i\frac{\eta}{2}\right). \tag{63}$$

For the tilted Néel state one has [120–122, 126]

$$\eta_1(\lambda) = -1 + \frac{T_1\left(\lambda + i\frac{\eta}{2}\right)}{\phi\left(\lambda + i\frac{\eta}{2}\right)}\frac{T_1\left(\lambda - i\frac{\eta}{2}\right)}{\bar{\phi}\left(\lambda - i\frac{\eta}{2}\right)}, \tag{64}$$

$$\rho_1^{(h)}(\lambda) = \frac{\sinh(\eta)}{\pi\left[\cosh(\eta) - \cos(2\lambda)\right]} - X_1\left(\lambda + i\frac{\eta}{2}\right) - X_1\left(\lambda - i\frac{\eta}{2}\right), \tag{65}$$

where now one has

$$\begin{aligned}T_1(\lambda) = -\frac{1}{8}\cot(\lambda)\Big[&8\cosh(\eta)\sin^2(\vartheta)\sin^2(\lambda) - 4\cosh(2\eta)\\&+(\cos(2\vartheta) + 3)(2\cos(2\lambda) - 1) + 2\sin^2(\vartheta)\cos(4\lambda)\Big],\end{aligned} \tag{66}$$

$$\phi(\lambda) = \frac{1}{8}\sin(2\lambda + i\eta)\left[2\sin^2(\vartheta)\cos(2\lambda - i\eta) + \cos(2\vartheta) + 3\right], \tag{67}$$

$$\bar{\phi}(\lambda) = \frac{1}{8}\sin(2\lambda - i\eta)\left[2\sin^2(\vartheta)\cos(2\lambda + i\eta) + \cos(2\vartheta) + 3\right], \tag{68}$$

and

$$\begin{aligned}X_1(\lambda) = &-(4\sinh(\eta)\sin^2(\vartheta)\cos(2\lambda) + \sinh(2\eta)(\cos(2\vartheta) + 3))\\&\times\Big(2\pi\big[8\cosh(\eta)\sin^2(\vartheta)\sin^2(\lambda) - 4\cosh(2\eta) + (\cos(2\vartheta) + 3)(2\cos(2\lambda) - 1)\\&\qquad\qquad + 2\sin^2(\vartheta)\cos(4\lambda)\big]\Big)^{-1}.\end{aligned} \tag{69}$$

Finally, for the Majumdar-Ghosh state one has [123]

$$\eta_1 = \frac{\cos(4\lambda) - 2\cosh(2\eta)}{\cos^2(\lambda)(\cos(2\lambda) - \cosh(2\eta))} - 1, \tag{70}$$

$$\rho_1^{(h)} = \frac{\sinh(\eta)}{\pi\left[\cosh(\eta) - \cos(2\lambda)\right]} - X_1\left(\lambda + i\frac{\eta}{2}\right) - X_1\left(\lambda - i\frac{\eta}{2}\right), \tag{71}$$

where

$$X_1 = \sinh(\eta)\frac{4\cos(2\lambda)(\sinh^2(\eta) - \cosh(\eta)) + \cosh(\eta) + 2\cosh(2\eta) + 3\cosh(3\eta) - 2}{8\pi(\cosh(2\eta) - \cos^2(2\lambda))}. \tag{72}$$

Having explicit expressions for all the root densities, we are ready to calculate the conjecture (4) for all these quenches. The functions $s_n(\lambda)$ are just straightforwardly obtained from the Yang-Yang entropy (43). For the velocity instead we have to solve numerically the set of coupled integral equations (47). The numerical results for the group velocities $v_n$ for several quenches in the XXZ chain are reported in Figure 3. The Figure shows the group velocities for a quench in the XXZ chain with $\Delta = 2$ plotted as a function of rapidity $\lambda$. Panels in different rows are for quenches from different initial states. Only results for string index $n \leq 3$ (panels on different columns) are shown. For all considered quenches and for all values of $\lambda$, $v_n$ decreases with the string index $n$. Interestingly, the maximum velocity is $v_M \approx 2$ for both the quenches from the Néel state and the dimer state, whereas it is $v_M \approx 1$ for the quench from the tilted ferromagnet. In the limit $\Delta \to \infty$ the solutions of the system (47) can be obtained analytically as a power series in $1/\Delta$ (see [10] for some analytical results).

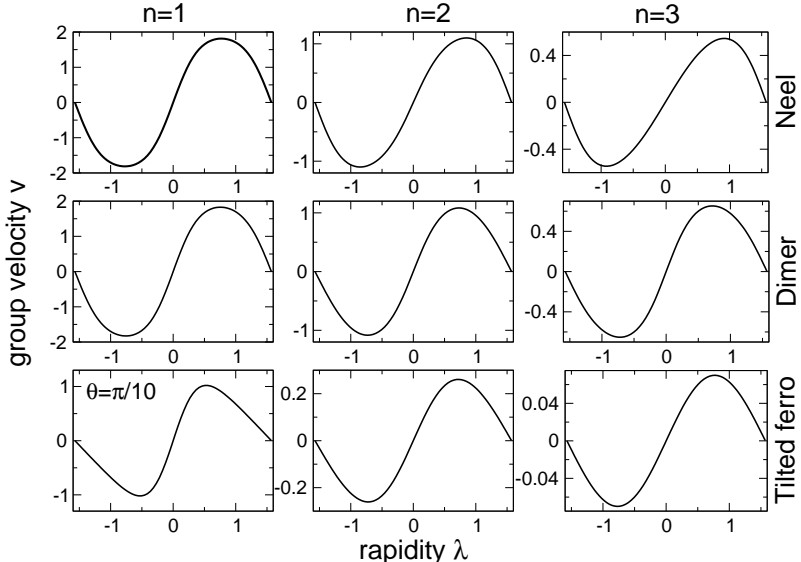

Figure 3: Group velocities of the low-lying excitations around the steady-state after a quench in the XXZ chain. All results are for chain anisotropy $\Delta = 2$. Group velocities are plotted against rapidity $\lambda$. Panels on different rows are for quenches from different initial states, namely the Néel state, the dimer state, and the tilted ferromagnet ($\vartheta$ is the tilting angle). Different rows correspond to bound states of different sizes (strings) $n = 1, 2, 3$. For all quenches the maximum velocity is obtained for $n = 1$ and the group velocity typically decreases upon increasing $n$.

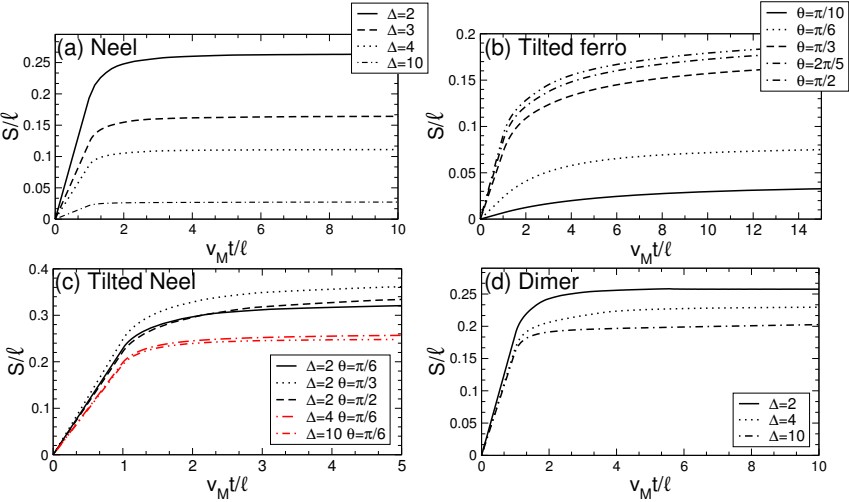

Figure 4: Quasiparticle prediction for the entanglement dynamics after a global quench in the XXZ chain. In all panels the entanglement entropy density $S/\ell$ is plotted against the rescaled time $v_M t/\ell$, with $\ell$ the size of $A$ and $v_M$ the maximum velocity. Different panels correspond to different initial states, namely the Néel state (a), tilted ferromagnet (b), tilted Néel (c), and dimer state (d). Different curves correspond to different values of the chain anisotropy $\Delta > 1$ and tilting angles $\vartheta$.

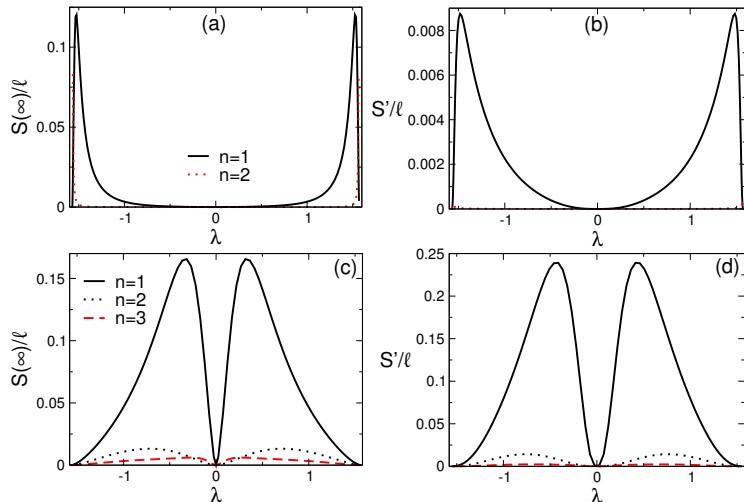

Figure 5: Quasiparticle contribution to the stationary entanglement entropy density $S(t = \infty)/\ell$ and to the entanglement production rate $S'(t)/\ell$ as function of the quasiparticle rapidity. In all panels the different curves correspond to bound states (strings) of different size $n = 1, 2, 3$. Panels (a)(b) show the results for the quench from the tilted ferromagnet with tilting angle $\vartheta = \pi/10$ and chain anisotropy $\Delta = 2$. Notice in both cases the peaks at $\lambda \approx \pm \pi/2$, which signal a large contribution of the slow quasiparticle to the entanglement dynamics. Panels (c)(d) show results for the quench from the Néel state in which the largest contributions correspond to $\lambda$ with small, but non-zero, value. Similar results are obtained for the quench from the dimer state and the tilted Néel. The contribution of the bound states with $n > 1$ are always much smaller than that for $n = 1$.

## 4.3 Entanglement dynamics

Let us repeat here the quasiparticle prediction (4) for the entanglement dynamics

$$S(t) = \sum_n \Big[ 2t \int_{2|v_n|t < \ell} d\lambda \, v_n(\lambda) s_n(\lambda) + \ell \int_{2|v_n|t > \ell} d\lambda \, s_n(\lambda) \Big], \tag{73}$$

where the sum is over the quasiparticle families $n$ (strings of different length), $v_n(\lambda)$ is the velocity of the entangling quasiparticles numerically calculated above, and $s_n(\lambda)$ denotes the contribution of each quasiparticle to the Yang-Yang entropy of the steady state in Eq. (43).

The exact numerical results obtained using (73) are illustrated in Figure 4. The different panels are for quenches from different initial states in the XXZ chain and several values of $\Delta$. For the quenches from the tilted Néel and ferromagnetic states $\vartheta$ is the tilting angle. In all panels the entropy density $S/\ell$ is plotted versus the rescaled time $v_M t/\ell$ with $v_M$ the maximum velocity, which is extracted from the Bethe ansatz. In all panels the expected behaviour with a linear increase at short times followed by an asymptotic saturation is observed. Interestingly, for all the quenches the larger steady-state entanglement is obtained for the smaller $\Delta$. The largest amount of entanglement is produced in the quench from the tilted Néel state (panel (c)). For the Néel quench the entropy vanishes in the limit $\Delta \to \infty$, which follows from the fact that the Néel state is the ground state of the XXZ chain in that limit, whereas it is finite for all other initial states. Finally, as already noticed in [10], for the quench from the tilted ferromagnet (see panel (b)) the linear regime seems to extend for $v_M t/\ell > 1$. However, the true linear regime extends only up to $v_M t/\ell = 1$. The behaviour observed in panel (b) is due to the large contributions to the entanglement entropy of slow quasiparticles (see [10]).

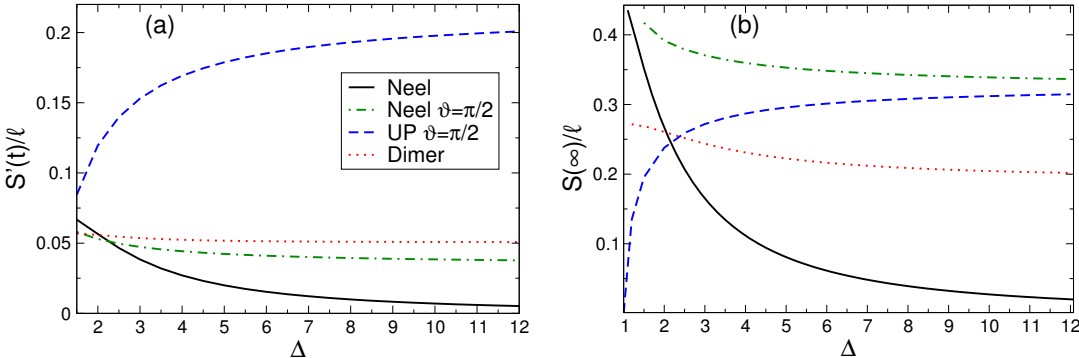

Figure 6: Anisotropy dependence of the entanglement after a global quench in the XXZ chains. Panel (a). The entanglement production rate $S'(t)/\ell$ as function of $\Delta$. Different curves correspond to different initial states. Panel (b). Steady-state entanglement entropy density $S(t = \infty)/\ell$ after the quench. The entanglement is identically zero in the limit $\Delta \to \infty$ for the quench from the Néel state and for $\Delta \to 1$ for the quench from the tilted ferromagnet.

A lot of important information is extracted by looking at the contribution to the entanglement dynamics of the individual quasiparticles. This is investigated in Figure 5 focusing on the steady-state entropy density $S/\ell$ (panels (a,c)) and on the slope of the linear growth at short times (b,d) $S'/\ell$ (we denote with *entanglement production rate* the quantity $S'(t)/\ell$ for $t < \ell/(2v_M)$ when it does not depend on time). Both quantities are plotted against the quasiparticles rapidity $\lambda$. All the results are for the quench in the XXZ chain with $\Delta = 2$. Panels (a) and (b) are for the quench from the tilted ferromagnet (with tilting angle $\vartheta = \pi/10$). Remarkably, the largest contribution to the steady-state entropy and to the entanglement production rate is in the region with large $\lambda$, which correspond to slow quasiparticles (see Figure 3). Also, the largest contribution is in the sector with $n = 1$ (continuous line in the Figure). The contributions of higher strings are negligible (the dotted line in Figure 5 (a) (b) is the contribution of the two-particle bound states). A striking different behaviour is observed for the quench from the Néel state (panels (c) and (d) in the Figure); now the largest contribution to the stationary entanglement and to the entanglement production rate is the region with small (but non-zero) rapidities, corresponding to fast quasiparticles. Similar to the quench from the tilted ferromagnet, the bound state contribution to the entanglement dynamics decays rapidly with their size.

Finally, we discuss the dependence of the stationary entropy and of the entanglement production rate on the chain anisotropy $\Delta$. This is shown in Figure 6. Clearly, for the quench from the Néel state the entropy is vanishing in the limit $\Delta \to \infty$, as already discussed. On the other hand, it remains finite for all the other quenches. Moreover, for the quench from the Néel state and the dimer state, both the steady-state entropy and the entanglement production rates exhibit their maximum value for $\Delta \approx 1$. In contrast, they vanish in the limit $\Delta \to 1$ for the quench from the tilted ferromagnet. This is expected because at $\Delta = 1$ the tilted ferromagnet becomes an eigenstate of the XXZ chain for any tilting angle. The large $\Delta$ behaviour can be understood analytically using perturbative methods. Here we discuss the behaviour of the steady-state entropy, although similar results can be derived for the entanglement production rate. It is straightforward to show that for the quench from the Néel state, in the limit $\Delta \to \infty$, the stationary entropy is

$$\frac{S}{\ell} = \frac{\ln \Delta^2}{\Delta} + o(\Delta^{-2}). \tag{74}$$

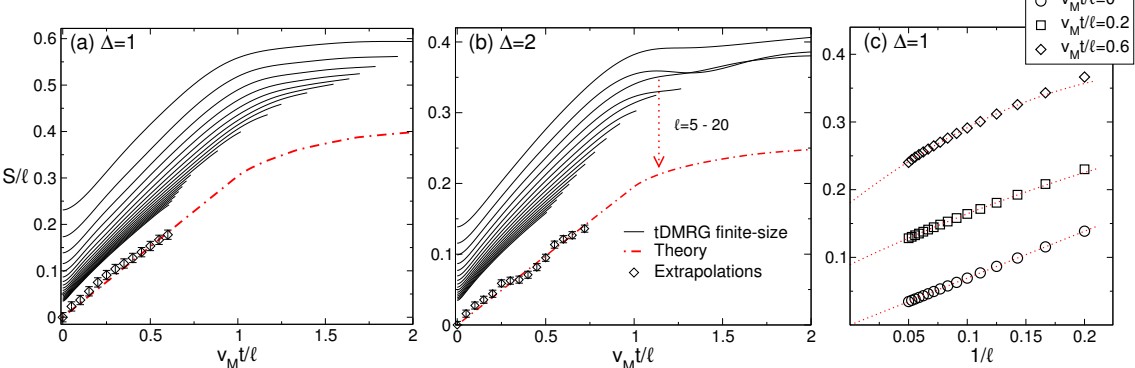

Figure 7: Post-quench dynamics of the von Neumann entanglement entropy in the XXZ spin chain: Comparison with tDMRG results. Here the entanglement entropy density $S/\ell$, with $\ell$ the subsystem size, is plotted against the rescaled time $v_M t/\ell$, with $v_M$ being the maximum velocity in the system. All the results are for the quench from the Néel state. Panel (a). Results for $\Delta = 1$. Continuous lines are tDMRG results for a chain with $L = 40$. Different lines correspond to different block sizes $\ell = 5-20$. The dashed line is the Bethe ansatz result in the scaling limit $t, \ell \to \infty$ with $x/t$ fixed. The diamonds are the numerical extrapolations (see panel (c)) in the thermodynamic limit. Panel (b). The same as in (a) for $\Delta = 2$. Panel (c). Numerical extrapolations of the tDMRG results in (a) in the thermodynamic limit. The panel plots $S/\ell$ versus $1/\ell$ for several values of $v_M t/\ell$ (different symbols). The curves are fits to $a + b/\ell + c/\ell^2$ with $a, b, c$ fitting parameters.

For the quench from the Majumdar-Ghosh state one has

$$\frac{S}{\ell} = -\frac{1}{2} + \ln 2 + o(\Delta^{-1}). \tag{75}$$

On the other hand, for the quenches from the tilted states the dependence of the root densities $\rho_n, \rho_n^{(h)}$ on the tilting angle is non-trivial even in the limit $\Delta \to \infty$, implying a non-trivial dependence for the entanglement entropy as well.

## 4.4 Numerical checks

In this section, using tDMRG simulations [130–132], we provide numerical evidence supporting our main result (73). Numerical results are presented in Figure 7. Panels (a) and (b) show tDMRG simulations for the quench from the Néel state in the XXZ chain at $\Delta = 1$ and $\Delta = 2$, respectively. The results in panel (b) are the same as in [10]. Both panels plot the entropy density $S/\ell$, with $\ell$ the size of subsystem $A$, as a function of the rescaled time $v_M t/\ell$, where $v_M$ is the maximum velocity calculated using the Bethe ansatz (see section 3.2). The continuous curves are tDMRG results for a chain with $L = 40$ sites and $\ell = 5-20$. The dashed-dotted line is the theoretical result (73) in the scaling limit. For both $\Delta = 1$ and $\Delta = 2$ scaling corrections are visible. The diamonds are extrapolations to the thermodynamic limit. These are obtained by fitting the data at fixed $v_M t/\ell$ to

$$\frac{S}{\ell} = s_\infty + \frac{a}{\ell} + \frac{b}{\ell^2}, \tag{76}$$

where $s_\infty, a, b$ are fitting parameters. The quality of the fits for the quench with $\Delta = 1$ (panel (a)) is illustrated in panel (c), plotting $S/\ell$ at fixed values of $v_M t/\ell$ (different symbols) as function of $1/\ell$. The dotted lines are fits to (76).

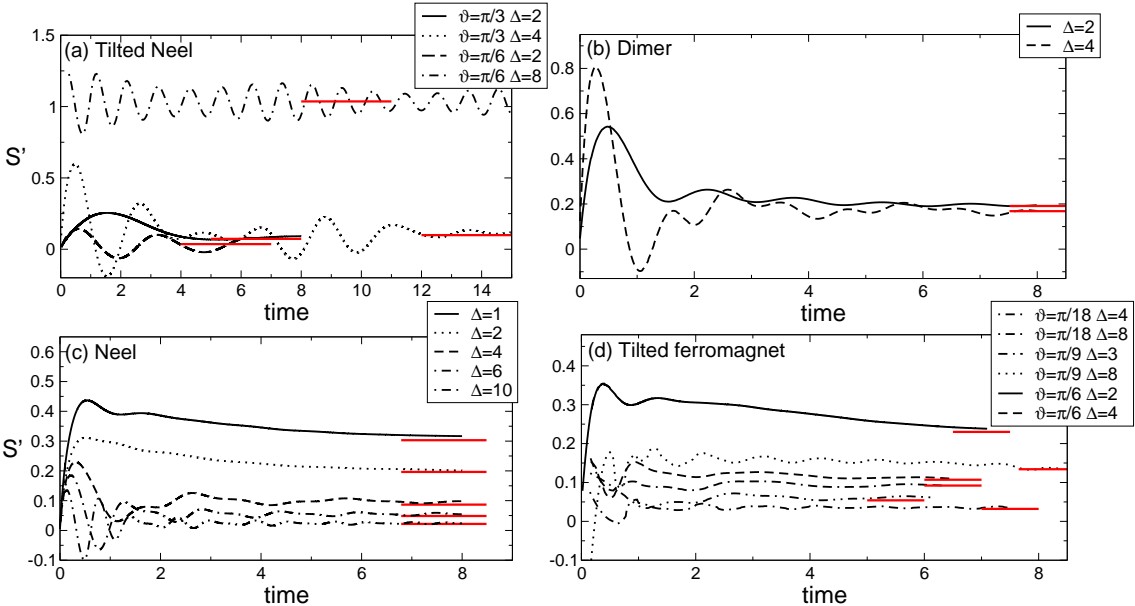

Figure 8: Entanglement production rate after a global quench in the XXZ spin chain. The panels plot $S'(t)$ as function of time. Different panels are for different initial states, namely the tilted Néel state (a), the dimer state (b), the Néel state (c), and the tilted ferromagnet (d). The curves are iTEBD numerical data for different anisotropy $\Delta$ and different tilting angles $\vartheta$. The horizontal segments are the predictions using the quasiparticle picture in the scaling limit.

We now turn to discuss further checks of (73) using the infinite Time-Evolving Block Decimation (iTEBD) [133] which works directly in the thermodynamic limit. Our results are discussed in Figure 8 (some results have been already reported in [10]). Different panels in the figure show the entanglement production rate $S'(t)$ plotted as a function of time for quenches with different initial states in the XXZ chain. The data shown in Figure 8 are the entanglement entropies for the half-infinite chain. Although no finite-size corrections are expected, finite-time corrections are visible in the Figure. The data exhibit a non-trivial dynamics at short times, often with oscillating behaviour. Interestingly, already at $t \approx 10$ for most of the quenches the data exhibit stationary behaviour. The horizontal lines in the Figure mark the quasiparticle prediction

$$S' = 2\sum_n \int_{-\pi/2}^{\pi/2} d\lambda\, v_n(\lambda) s_n(\lambda). \tag{77}$$

The agreement between (77) and the iTEBD data is spectacular for all the quenches. Note that in the vicinity of $\Delta = 1$ a slower relaxation to the stationary behaviour takes place, especially for the quenches from the Néel state and from the tilted ferromagnet: longer times would be needed in order to provide a more robust check of (77).

## 4.5  Mutual information

In this section we focus on the post-quench dynamics of the mutual information between two blocks. Considering the tripartition $A_1 \cup A_2 \cup B$ (with $A_1$ and $A_2$ two intervals of equal length $\ell$ and at distance $d$ and $B$ the rest of the chain), the von Neumann mutual information is defined as

$$I_{A_1:A_2} \equiv S_{A_1} + S_{A_2} - S_{A_1 \cup A_2}, \tag{78}$$

with $S_{A_{1(2)}}$ and $S_{A_1 \cup A_2}$ being the entanglement entropies of $A_{1(2)}$ and $A_1 \cup A_2$, respectively.

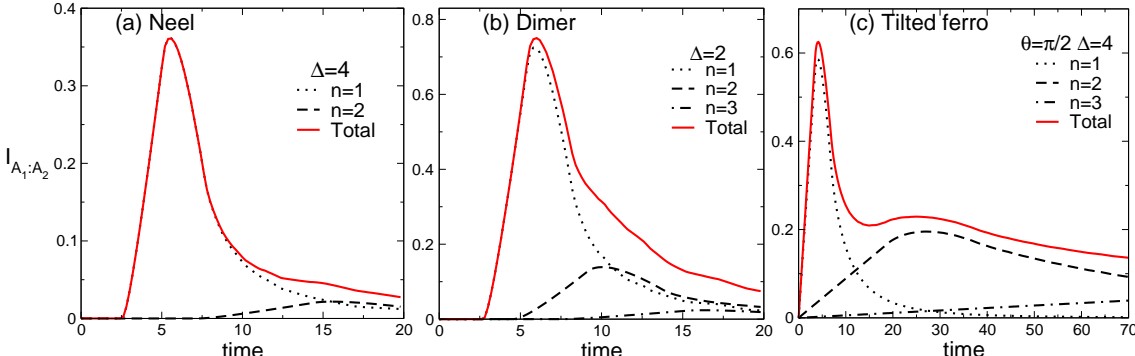

Figure 9: Post-quench dynamics of the mutual information $I_{A_1:A_2}$ between two intervals $A_1$ and $A_2$ after a quench in the XXZ chain. Panel (a) shows $I_{A_1:A_2}$ for the quench from the Néel state for $\Delta = 4$. Here $A_1$ and $A_2$ are two disjoint intervals of equal length $\ell = 10$ at distance $d = 10$ in units of the lattice spacing. Different curves correspond to the contributions of bound states of different size $n$. The continuous (red) line is obtained by summing over all the bound states. Panel (b) is the same as in (a) for the quench from the Majumdar-Ghosh state for the XXZ chain with $\Delta = 2$. Panel (c). Post-quench dynamics of $I_{A_1:A_2}$ for the quench from the tilted ferromagnet in the XXZ chain with $\Delta = 4$. Here $A_1$ and $A_2$ are two equal-length intervals with $\ell = 10$ at distance $d = 0$. Note the second peak at $t \approx 30$ resulting from the contribution of the two-particle bound states.

Using the quasiparticle picture, it is straightforward to derive a prediction for the mutual information. When only one type of quasiparticles is present with fixed group velocity $v$ (as in a conformal field theory), the prediction for the mutual information is simply obtained by counting the quasiparticles arriving to each interval, obtaining [6]

$$I_{A_1:A_2} \propto -2\max((d+\ell)/2, vt) + \max(d/2, vt) + \max((d+2\ell)/2, vt). \tag{79}$$

Formula (79) predicts $I_{A_1:A_2} = 0$ for $vt \leq d/2$, followed by a linear increase for $d/2 < vt \leq (d+\ell)/2$ and a linear decrease up to $vt = (d+2\ell)/2$. The first region corresponds to $A_1$ and $A_2$ being entangled with the environment $B$ but not mutually entangled. At time $t = d/(2v)$ quasiparticles originated at the same point in space start to connect $A_1$ and $A_2$. The linear increase up to $t = (d+\ell)/(2v)$ correspond to entangled quasiparticles traveling in the two subsystems. At time $t = (d+\ell)/(2v)$ the entangled quasiparticles start leaving the two subsystems. Finally, at $t = (d+2\ell)/(2v)$ there are no entangled quasiparticles connecting $A_1$ and $A_2$ and the mutual information vanishes again.

In the presence of different species of quasiparticles with different velocities, one has to integrate (79) over the full quasiparticle content to obtain

$$I_{A_1:A_2} = \sum_n \int d\lambda s_n(\lambda) \Big[ -2\max((d+2\ell)/2, v_n(\lambda)t)$$
$$+ \max(d/2, v_n(\lambda)t) + \max((d+4\ell)/2, v_n(\lambda)t) \Big], \tag{80}$$

which is valid for infinite systems. For a finite chain, (80) applies before the revival time.

The exact numerical results for $I_{A_1:A_2}$ obtained using (80) for quenches in the XXZ chain are shown in Figure 9. Panel (a) shows results for the quench from the Néel state in the XXZ chain with $\Delta = 4$. The result for $I_{A_1:A_2}$ (full line in the Figure) is for two disjoint intervals of equal length $\ell = 10$ at distance $d = 10$. Clearly, one has that for $d/(2v_M)$, with $v_M \approx 2$ the maximum velocity, the mutual information is zero. A linear behaviour is clearly visible at

larger times up to $(d+\ell)/(2v_M)$, where the mutual information reaches a maximum. A linear decrease is subsequently observed. Interestingly, the presence of slow quasiparticles leads to a slow decay of the mutual information at long times, instead of a sudden vanishing behaviour at $t=(d+2\ell)/(2v_M)$. A similar slow decay has been numerically observed in free bosonic models [83].

It is also interesting to investigate the effects of the bound states on the mutual information dynamics. The dotted and dashed lines in Figure 9 denote the contributions of the two-particle and three-particle bound states, respectively. Interestingly, the contributions of the bound states rapidly decay with their size. Moreover, the bound-state contributions are shifted at longer times, reflecting their smaller group velocities (see Figure 3). Similar qualitative results are observed for the quench from the dimer state (reported in Figure 9 (b)). Finally, Figure 9 shows results also for the quench from the tilted ferromagnet. The data are for $\Delta=4$ and tilting angle $\vartheta=\pi/2$. The results are for two adjacent equal-length intervals with $\ell=10$. In contrast with panels (a) and (b), an additional second peak is observed in the mutual information. As it is clear from the Figure, this is due to the contribution of the two-particle bound states (dashed line). This last result suggests that the mutual information, at least in some case, can be used to reveal the bound state content of integrable models. This idea has already been put forward in [100] during the study of quenches in the spin-1 Lai-Sutherland model.

# 5 Entanglement dynamics in the Lieb-Liniger model

In this section we provide exact results for the entanglement dynamics after the quench from the Bose-Einstein condensate (BEC) in the Lieb-Liniger model. We discuss both the Lieb-Liniger model with repulsive interactions, as well as with attractive ones. Quantum quenches in the Lieb-Liniger model have been the focus of intensive investigations [134–160] and here we will largely use the results from Refs. [140] and [155] for repulsive and attractive cases respectively. We should mention that, in contrast with the XXZ chain, here we cannot provide a numerical check of our preditions. This is due to the fact that as of now for models in the continuum there are no efficient numerical methods, such as tDMRG.

## 5.1 Lieb-Liniger model and its Bethe Ansatz solution

The Lieb-Liniger model consists of a system of $N$ interacting bosons on a ring of length $L$. The model is defined by the hamiltonian

$$H=-\frac{\hbar^2}{2m}\sum_{j=1}^{N}\frac{\partial^2}{\partial x_j^2}+2c\sum_{j<k}\delta(x_j-x_k), \tag{81}$$

where $m$ is the mass of the bosons and $c$ is the interaction strength. In the following we set $\hbar=2m=1$. In second quantisation (81) reads

$$H=\int_0^L dx\left\{\partial_x\Psi^\dagger(x)\partial_x\Psi(x)+c\Psi^\dagger(x)\Psi^\dagger(x)\Psi(x)\Psi(x)\right\}, \tag{82}$$

with $\Psi(x)$ bosonic fields satisfying the standard commutation relations $[\Psi(x),\Psi^\dagger(y)]=\delta(x-y)$. In the limit $c\to\infty$, (82) becomes equivalent to a system of hard-core bosons. For any value of $c$, the Lieb-Liniger model can be solved using Bethe ansatz [161]. In this work we consider both the repulsive regime with $c>0$, as well as the

attractive one with $c < 0$. We define $\bar{c} \equiv |c|$. We also introduce the dimensionless coupling $\gamma$ as

$$\gamma \equiv \frac{|c|}{D}, \quad \text{with } D \equiv \frac{N}{L}. \tag{83}$$

The Bethe equations for the Lieb-Liniger model are [110, 161]

$$\frac{2\pi I_j}{L} = \lambda_j + \text{sgn}(c)\frac{2}{L}\sum_{k=1}^{N} \arctan\left(\frac{\lambda_j - \lambda_k}{\bar{c}}\right). \tag{84}$$

The eigenstates energy $E$ and total momentum $P$ are given as

$$E = \sum_j \lambda_j^2, \qquad P = \sum_j \lambda_j = \frac{2\pi}{L}\sum_j I_j. \tag{85}$$

The structure of the solutions of the Bethe equations depends dramatically on the sign of the interactions. Specifically, for $c > 0$, i.e., for repulsive interactions, only real solutions of (84) are present. Consequently (84) is of the form (32) for the only species of particles after the straightforward identification of the various functions. In the thermodynamic limit the solutions of the BGT equations become dense on the real axis and, since there are no bound states, there is a single particle density $\rho$ and hole density $\rho^{(h)}$, with $\rho^{(t)} = \rho + \rho^{(h)}$ which is a major simplification compared to the standard case. The Bethe equations for these root densities (84) are

$$\frac{1}{2\pi} + \int_{-\infty}^{\infty} d\lambda' K(\lambda - \lambda')\rho(\lambda') = \rho^{(t)}(\lambda), \tag{86}$$

where the kernel $K$ is given as $K(\lambda) = c/[\pi(\lambda^2 + c^2)]$

For attractive interactions $c < 0$, the eigenstates of the model contain non-trivial multi-particle bound states that, as usual, can be understood with the string hypothesis, i.e. they have the form (31) with $\eta = \bar{c}$. The Bethe-Gaudin-Takahashi (BGT) equations for the Lieb-Liniger gas are of the form (32) with $\pi_n(\lambda) = n\lambda$ and elementary kernel $\theta_n(\lambda)$ given by [110]

$$\theta_n(\lambda) = 2\arctan\left(\frac{2\lambda}{n\bar{c}}\right). \tag{87}$$

For the attractive Lieb-Liniger the energy and momentum in (85) can be rewritten as (34) with

$$\epsilon_n(\lambda) = n\lambda^2 - \frac{c^2}{12}n(n^2 - 1). \tag{88}$$

In the thermodynamic limit, there are infinite particle densities $\{\rho_n\}_{n=1}^{\infty}$, hole densities $\{\rho_n^{(h)}\}_{n=1}^{\infty}$, and total densities $\{\rho_n^{(t)}\}_{n=1}^{\infty}$ as the sum of the other two. The thermodynamic version of the BGT equations (32) takes the explicit form

$$\frac{n}{2\pi} - \sum_{m=1}^{\infty} \int_{-\infty}^{\infty} d\lambda' K_{n,m}(\lambda - \lambda')\rho_m(\lambda') = \rho_n^{(t)}, \tag{89}$$

with

$$K_{n,m}(\lambda) = (1 - \delta_{n,m})a_{|n-m|}(\lambda) + 2a_{|n-m|+2}(\lambda) + \cdots + 2a_{n+m-2}(\lambda) + a_{n+m}(\lambda), \tag{90}$$

and

$$a_n(\lambda) = \frac{2}{\pi|c|n}\frac{1}{1 + (\frac{2\lambda}{n|c|})^2}. \tag{91}$$

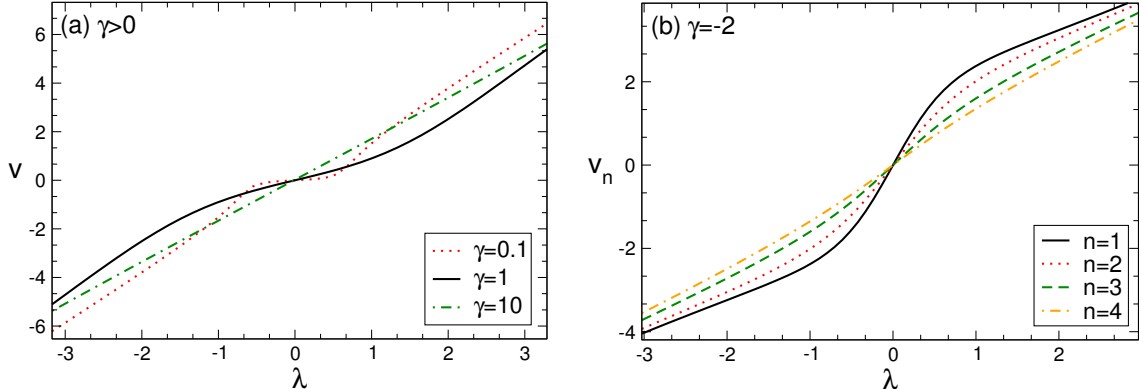

Figure 10: Group velocities of the low-lying excitations around the steady-state after the quench from the Bose condensate (BEC) in the Lieb-Liniger model. Panel (a). Results for the repulsive Lieb-Liniger. The different curves are the group velocities $v$ plotted as a function of the rapidity $\lambda$ for several values of the interaction strength $\gamma$. Panel (b) reports the group velocities for the attractive Lieb-Liniger model with $\gamma = -2$. The different curves are for the different bound states. Notice that in both cases, the velocities are unbounded and grow linearly as $\lambda \to \pm\infty$.

## 5.2 Quench from the Bose condensate

Here we briefly detail the TBA treatment for the quantum quench from the Bose condensate state in the Lieb-Liniger model. In the BEC the bosons are uniformly distributed in the interval $[0, L]$. The steady state arising at infinite time after the quench is fully described by a particular thermodynamic macrostate.

### 5.2.1 Repulsive case

The quench action solution for the quench in the repulsive Lieb-Liniger has been provided in [140]. The thermodynamic macrostate describing the post-quench steady-state is identified by the densities $\rho(\lambda), \eta(\lambda)$ [140]:

$$\rho(\lambda) = \frac{1}{2\pi} \frac{\tau}{2} \frac{da(\lambda/c)}{d\tau}, \qquad \eta(\lambda) = \frac{1}{a(\lambda/c)}, \tag{92}$$

written in terms of the the auxiliary function

$$a(\lambda) \equiv \frac{2\pi\tau}{\lambda \sinh(2\pi\lambda)} I_{1-2i\lambda}(4\sqrt{\tau}) I_{1+2I\lambda}(4\sqrt{\tau}). \tag{93}$$

Here $\tau = 1/\gamma$ and $I_\alpha(x)$ are the modified Bessel functions of the first kind.

The calculation of the group velocities of the low-lying excitations around the macrostate that describes the steady-state follows the general derivation of section 3.2 with the major simplification of having a single set of rapidities. Figure 10 (a) shows numerical results for the group velocities of the low-lying excitations around the post-quench steady state for several values of the interaction strength $\gamma$ (different curves in the Figure). At large $|\lambda|$ the interactions are negligible and the linear behaviour $v \propto 2\lambda$ is observed, reflecting the "free" dispersion $E = \lambda^2$ and the absence of a maximum velocity. We anticipate that this fact will have striking consequences in the behaviour of the entanglement entropy (see 5.3).

### 5.2.2 Attractive case

We now consider the quench from the Bose condensate in the attractive gas for which the thermodynamic macrostate describing the steady state is identified by the set of densities $\{\rho_n\}_{n=1}^{\infty}$ and $\{\eta_n\}_{n=1}^{\infty}$. The solution for this problem has been provided in [155]. The densities $\eta_n$ satisfy the recursion relations [155]

$$\eta_n(x) = \frac{\eta_{n-1}(x - \frac{i}{2})\eta_{n+1}(x + \frac{i}{2})}{1 + \eta_{n-2}(x)} - 1, \tag{94}$$

with $x \equiv \lambda/c$ and

$$\eta_1(x) = \frac{x^2(1 + 4\tau + 12\tau^2 + (5 + 16\tau)x^2 + 4x^4)}{4\tau^2(1 + x^2)}. \tag{95}$$

The particle densities $\rho_n(x)$ are [155]

$$\rho_n(x) = \frac{\tau}{4\pi} \frac{1}{1 + \eta_n(x)} \frac{d\, 1/\eta_n(x)}{d\tau}. \tag{96}$$

The group velocities of the low-lying excitations around the macrostate describing the steady-state can be calculated following the general derivation of Sec. 3.2. Figure 10 (b) shows numerical results for these group velocities $v_n$ for the different multi-particle bound states as a function of $\lambda$. The results are for fixed $\gamma = -2$. As for the repulsive case, at large momenta, the interactions are negligible and the free-like behaviour $v \propto 2\lambda$ is found.

### 5.3 Entanglement dynamics in the Lieb-Liniger model

We now turn to discuss the post-quench dynamics of the entanglement entropy for the repulsive Lieb-Liniger model as given by the quasiparticle prediction (4). In the present case, (4) greatly simplifies because of the presence of a single species of quasiparticles and it can be written as

$$S(t) = 2t \int_{2|v|t<\ell} d\lambda\, v(\lambda) s(\lambda) + \ell \int_{2|v|t>\ell} d\lambda\, s(\lambda), \tag{97}$$

where $v(\lambda)$ is the group velocity of the entangling quasiparticles of the previous section, and $s(\lambda)$ is the thermodynamic Yang-Yang entropy of the steady state.

The dynamics of the entanglement entropy obtained from (97) is shown in Figure 11. Panel (a) in the Figure plots the entropy density $S/\ell$ versus the rescaled time $t/\ell$. The different curves in the Figure correspond to different values of the repulsive interaction strength. Interestingly, for all values of $\gamma$ the entropy exhibits a non-linear growth with time, even at short times, and it saturates at asymptotically long times. The non linear behaviour at short times is due to the absence of a maximum velocity (see Figure 10). The almost linear behaviour of the entanglement entropy for small values of $\gamma$ is due to the very low weight of fast quasiparticles, as it should be clear from Fig. 10. Anyhow, at a closer analysis, a strictly linear behaviour never takes place for any value of $\gamma$. The maximum stationary entanglement entropy is obtained in the limit $\gamma \to \infty$, when the system is equivalent to a system of hard-core bosons. In this limit, we find $S/\ell = 2$, as already known [73]. In order to understand the saturation behaviour at long times it is useful to investigate the quasiparticle contribution to the steady-state entanglement entropy. This is reported in Figure 11 (b) which shows the entropy density $S/\ell$ contribution versus the quasiparticle rapidity $\lambda$. The different curves correspond to different values of the interaction strength. For all values of $\gamma$ the quasiparticles contributions decay rapidly as $\lambda \to \infty$. Upon increasing $\gamma$, quasiparticles with larger rapidity

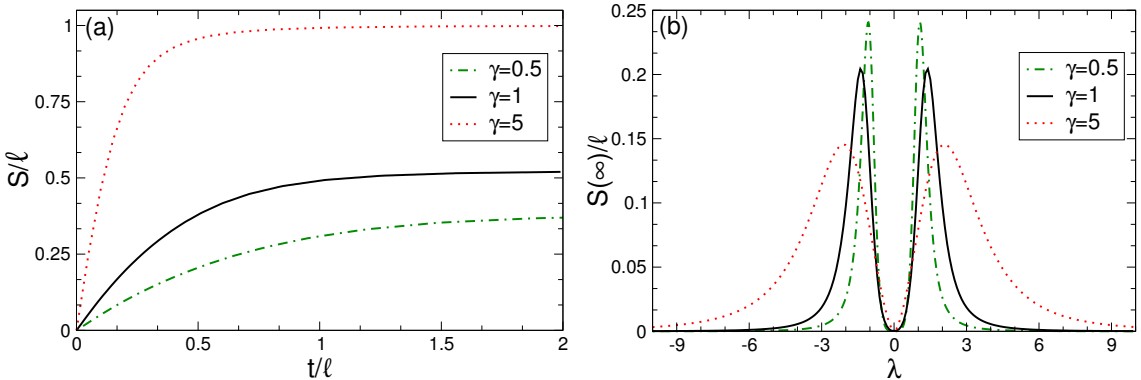

Figure 11: Entanglement entropy dynamics in the repulsive Lieb-Liniger model after the quench from the Bose condensate (BEC). (a) Quasiparticle picture prediction for $S/\ell$ plotted versus the rescaled time $t/\ell$. The different curves correspond to different values of the interaction strength $\gamma$. Notice the absence of the linear regime at short times. (b). Contributions of the quasiparticles of rapidity $\lambda$ to the stationary entanglement. The different curves are for different values of $\gamma$ (same as in (a)).

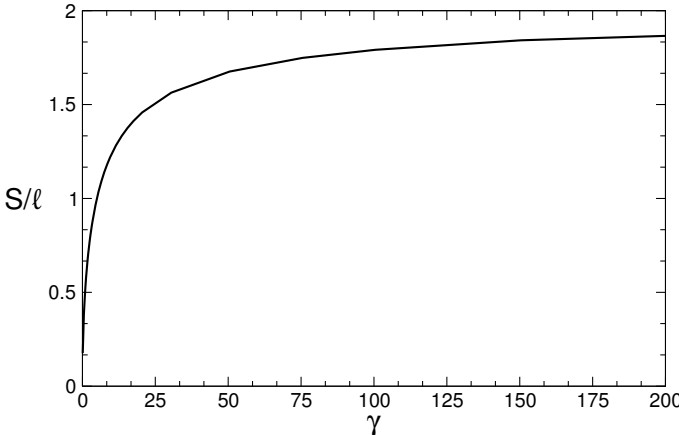

Figure 12: Steady-state entanglement entropy density $S/\ell$ after the quench from the Bose condensate in the repulsive Lieb-Liniger model as function of the interaction strength $\gamma$. For infinite repulsion $\gamma \to \infty$, the result $S/\ell = 2$ for hard-core bosons is recovered.

contribute more significantly to the steady-state entropy, which is one of the factors explaining why the stationary entropy increases with $\gamma$. This is better shown in Figure 12 that reports the entropy density $S/\ell$ versus $\gamma$. The entropy density monotonically increases with $\gamma$ and it vanishes for $\gamma \to 0$, i.e. in the absence of a quench. In the limit $\gamma \to \infty$ the result $S/\ell = 2$ [73] for hard-core bosons is recovered.

We now turn to discuss the entanglement dynamics after the quench from the Bose condensate in the attractive Lieb-Liniger model. In this case, all the multi-boson bound states contribute to the entanglement which is then described by (4) that we repeat here for convenience:

$$S(t) = \sum_{n} \Big[ 2t \int\limits_{2|v_n|t<\ell} d\lambda \, v_n(\lambda) s_n(\lambda) + \ell \int\limits_{2|v_n|t>\ell} d\lambda \, s_n(\lambda) \Big]. \tag{98}$$

Numerical results for the entanglement evolution obtained using (98) are shown in Figure 13. Panel (a) shows results for $S/\ell$ for the quench with $\gamma = -2$ plotted as a function of the rescaled time $t/\ell$. The different curves in the panel are the entanglement entropies in which the differ-

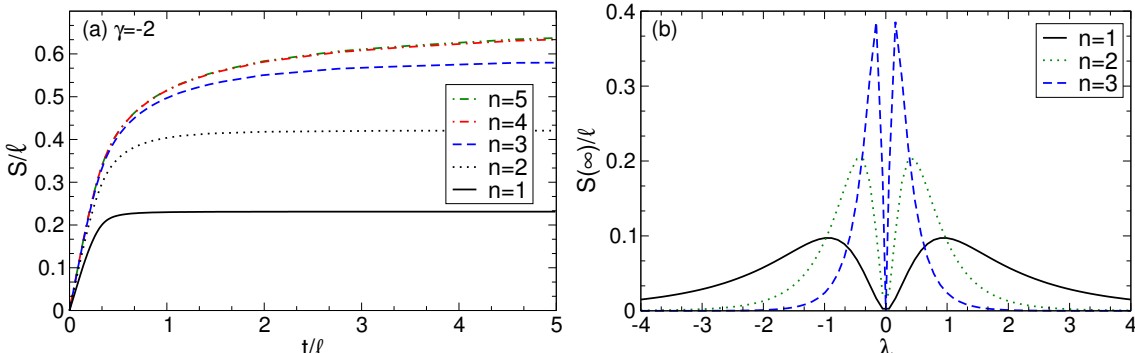

Figure 13: Entanglement dynamics after the quench from the Bose condensate in the attractive Lieb-Liniger model. Panel (a) shows the entropy density $S/\ell$ plotted versus the rescaled time $t/\ell$. The curves are Bethe ansatz results for fixed interaction strength $\gamma = -2$ in which all the bound states with size up to $n$ have been included in the sum (98). Panel (b). Contributions of the different bound states to the steady-state entanglement entropy. $S/\ell$ is plotted as a function of the quasiparticle rapidity $\lambda$. Different lines correspond to different bound-state sizes $n$.

ent multi-boson bound states up to size $n$ have been taken into account in the sum (98). Only results for $n \leq 5$ are shown. We verified that for this value of $\gamma$ and *in the time window reported in the plot*, the contributions of the bound states with $n > 5$ are negligible. As for the repulsive case (cf. Figure 11) there is no linear increase in the short time regime. The contribution of bound states with different rapidity is investigated in Figure 13 (b) plotting $S/\ell$ as a function of rapidity $\lambda$ for different values of $n$. Interestingly, the maximum contribution of the bound states increases with their size, although the support of $S/\ell$ as a function of $\lambda$ shrinks with increasing $n$. As a consequence of this very peculiar velocity distribution, we have that the larger bound states have a dominant velocity that is smaller and smaller as $n$ increases. Thus their effect will manifest at longer times. This is already clear from the panel (a) in Fig. 13 where we can notice that the contributions with $n = 3, 4, 5$ have a visible effect some time after the quench. Consequently, we expect that larger bound states can have non-negligible contributions at some larger time not displayed in the figure.

We turn now to discuss the steady-state entropy as a function of the interaction strength. Clearly, the entropy density increases with $\gamma$, similar to the repulsive case (see Figure 13). The behaviour in the limit $\gamma \to \infty$ can be understood analytically. In the limit $\gamma \to \infty$, one has that the support of the root densities $\rho_n(\lambda)$ and $\rho_n^{(h)}(\lambda)$ shrinks around $\lambda = 0$. Specifically, in the limit $\tau \to 0$ one has that

$$\rho_1(x) \approx \frac{2\tau^2}{\pi(x^2 + 4\tau^2)}, \quad \rho_1^{(h)}(x) \approx \frac{x^2}{2\pi(x^2 + 4\tau^2)}, \tag{99}$$

$$\rho_2(x) \approx \frac{16\tau^4}{\pi(x^2 + 16\tau^4)}, \quad \rho_2^{(h)}(x) \approx \frac{x^2}{\pi(x^2 + 16\tau^4)}, \tag{100}$$

$$\rho_3(x) \approx \frac{96\tau^6}{\pi(9x^2 + 64\tau^6)}, \quad \rho_3^{(h)}(x) \approx \frac{27x^2}{2\pi(9x^2 + 64\tau^6)}, \tag{101}$$

$$\rho_4(x) \approx \frac{128\tau^8}{\pi(81x^2 + 64\tau^8)}, \quad \rho_4^{(h)}(x) \approx \frac{162x^2}{\pi(81x^2 + 64\tau^8)}, \tag{102}$$

where $x \equiv \lambda/\bar{c}$. Interestingly, Eq. (99) implies that

$$\rho_1^{(t)} \approx \frac{1}{2\pi}, \tag{103}$$

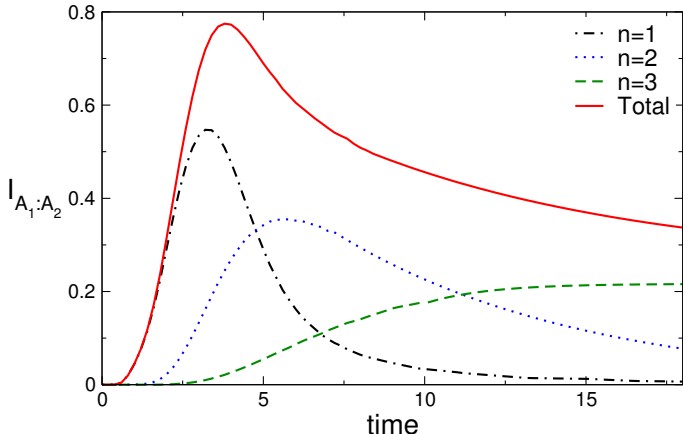

Figure 14: Non-equilibrium dynamics of the mutual information $I_{A_1:A_2}$ between two disjoint intervals $A_1$ and $A_2$ after the quench from the BEC to the attractive Lieb-Liniger with $\gamma = -2$. $I_{A_1:A_2}$ (continuous line) is plotted as function of the time after the quench. Results are for two intervals of length $\ell = 10$ at distance $d = 10$. The contributions of the different multiparticle bound states of different sizes $n$ are also reported.

i.e. that in the limit of infinite attractive interaction the quasiparticles with $n = 1$ behave as free fermions. For generic $n$, the total density $\rho_n^{(t)}$ is consistent with the ansatz

$$\rho_n^{(t)}(\lambda) = \frac{n}{2\pi}. \tag{104}$$

Crucially, from (99)-(102) it is clear that the support of the higher densities $\rho_n$ and $\rho_n^{(h)}$ for $n > 1$ shrinks faster, i.e., with a higher power of $\tau$, in the limit $\tau \to 0$, implying that the multi boson bound states do not contribute to the leading behaviour of the steady-state entanglement entropy. Also, we should remark that, although the functional form of the densities in the limit $\gamma \to \infty$ appear to be simple, we were not able to generalize the results (99)-(102) to arbitrary $n$.

We can derive the average energy and particle density using (99)-(102). The boson density in the limit $\gamma \to \infty$ is determined by the strings with $n = 1$ and it is given as

$$|c| \int_{-\infty}^{\infty} dx \frac{2\tau^2}{\pi(x^2 + 4\tau^2)} = D, \tag{105}$$

as it should. Using (99)-(102), it is straightforward to check that the contributions of the bound states are vanishing as $\propto \tau^{n-1}$. On the other hand, for the energy density the contribution of each bound state diverges in the limit $\gamma \to \infty$ as expected because the energy of the post-quench hamiltonian calculated on the BEC state diverges as $\gamma \to \infty$.

Using (103), (99), and (100) in the definition of the Yang-Yang entropy, the stationary entanglement in the limit $\tau \to 0$ is determined by the strings with $n = 1$, and it is given as

$$S = 2\ell + o(\ell). \tag{106}$$

Interestingly, Eq. (106) is the same as for the BEC quench in the repulsive Lieb-Liniger [73].

### 5.3.1 Mutual information

Finally, we investigate the behaviour of the mutual information between two intervals. The quasiparticle formula for the mutual information is the same as that for the XXZ chain (80).

Figure 14 shows $I_{A_1:A_2}$ for two disjoint intervals with equal length $\ell = 10$ at distance $d = 10$ for the Lieb-Liniger gas with $\gamma = -2$. The continuous line denotes $I_{A_1:A_2}$ while the other curves are the individual contributions of the bound states with $n = 1, 2, 3$. Interestingly, the mutual information exhibits a peak at short times, which is followed by a quite slow vanishing behaviour as $t \to \infty$. This slow relaxation is due to the significant contributions of the multi-bosons bound states. For all the bound states, a peak is observed at relatively short times, followed by a vanishing behaviour at long times. However, the position of the peak is shifted to longer times for the larger bound states.

## 6    Conclusions

In this paper we provided a thorough analysis of the framework put forward in [10] for the time evolution of the entanglement entropy which combines the quasiparticle picture of [6] with the exact knowledge of the stationary state coming from integrability. This approach is expected to hold in generic one-dimensional integrable systems. Here, we provided predictions, valid under rather general conditions, for arbitrary free systems, both bosonic and fermionic. These results have been tested against exact computations for the Ising and the harmonic chains. We also provided new results for the Heisenberg anisotropic spin chain (XXZ chain), which was the only model analysed in [10]. We finally derived theoretical predictions for the entanglement dynamics in the Lieb-Liniger model which have not been checked against numerical simulations, although it would be very interesting to do so. Specifically, it would be useful to verify the non-linear behaviour of the entropy at short times. A promising direction to perform this check is to extend the framework of continuous matrix product states [162] to simulate non-equilibrium systems. Alternatively, one could study the non-equilibrium dynamics of a very dilute Bose Hubbard model (as done in [163]), but this is computationally demanding.

A crucial observation is that Eq. (4) has been conjectured on the basis that the initial state acts as a source of *pair* of quasiparticle excitations with opposite momentum. In Bethe ansatz language, this assumption reflects the property that only *parity-invariant eigenstates* (as defined in [137, 140]) have non zero-overlap with the initial state. Recently, there is a broad consensus emerging about the idea that only quenches from these initial states are exactly solvable for genuinely interacting integrable models, as first proposed in the context of quantum field theory [164] and later for lattice integrable models [165]. However, states with non-zero overlap with generic eigenstates do exist and it is fundamental to understand how (4) generalises. In this respect, free models can be a useful playground because they can be solved even relaxing this assumption. Examples of exact results for quenches from non parity invariant states have been provided recently for the Hubbard chain with infinite repulsion [103] (which is mappable to free fermions), and the entanglement dynamics can be described by a suitable generalisation of (3) [104].

A main open problem is the generalisation of the approach of this paper to Rényi entanglement entropies. While in Refs. [166–168] it has been shown how to derive analytically the stationary value of these entanglement monotones, a complete quasiparticle description for their full-time evolution is still lacking. On the same line of thoughts, it would be important to provide a semiclassical picture for more complex entanglement measures, such as the negativity [169–171], which quantify the entanglement also in mixed states. In this respect, a promising direction is to study the dynamics of the negativity in the harmonic chain, for which exact calculations are possible [83].

# Acknowledgments

V.A. acknowledges support from the European Union's Horizon 2020 under the Marie Sklodowoska-Curie grant agreement No 702612 OEMBS.

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
