# Peer review of "Entanglement dynamics after quantum quenches in generic integrable systems"

_SciPost Physics, doi:SciPost Phys. 4, 017 (2018)_

## Round 1 · Referee Report · Anonymous (Referee 1) · 2018-2-14

Strengths

1- Analytical results on a timely subject. 2- The numerical verifications are rather convincing. 3- Easy to read.

Weaknesses

1- Most of the results rely heavily on another paper, Ref[10], by the same authors.

Report

In this paper, the two authors study the large time entanglement dynamics following a quantum quench in several Bethe Ansatz integrable models. Explicit analytical formulas are derived in various quench setups. Those are then verified numerically using the DMRG algorithm.

The starting point is equation (4), which was derived in Ref.[10] by the same authors. The result is then applied to several interesting quench situations, where it can be used to derive explicit analytical formulas. Most of those protocols were considered before by other authors, but the entanglement results are new.

Overall the paper is interesting and timely. The derivations are explained in great detail, which makes the manuscript rather easy to follow. Given the interest in entanglement growth and effective descriptions of quantum systems out of equilibrium, the results presented here will be quite useful to researchers working in this field.

For these reasons I recommend publication, provided the authors address the minors points listed below.

Requested changes

a) The logic in the introduction could be improved a bit, I think. The authors spend a great deal of time discussing equation (3), and where it comes from. Then, suddenly, they switch to (4), a multi-species generalization with very little explanation. It should be at least mentioned, already around (4) that (i) in general there are more than one species (ii) the different species do not interact.

Another confusing aspect is the equality in (3), (4). It is not clear whether those are meant as strict equalities, or as an asymptotic expansion as in (106). It seems to me there could be order one corrections to (3) and (4). Those should be mentioned at the beginning, to avoid confusion.

b) Pages 2-3. The sentence "Remarkably, in recent years [...]" seems a little bit out of place, and should perhaps appear earlier in the introduction.

c) Pages 5-6, section 2.1.1. Mention that periodic boundary conditions are assumed.

d) In section 3.1, it's not completely clear starting from where $\Delta>1$ is assumed. Could the author also comment somewhere on the more complicated "gapless" case?

e) The expression "quench action" appears for the first time in section 5. The terminology used in sections 4 and 5 should perhaps be made more consistent, however the authors see fit.

f) Page 27. This is mentioned in the conclusion, but it might be useful to add a quick comment regarding the difficulties in numerical simulations in the Lieb-Liniger model.

Typos, etc.

a) Page 1, second paragraph. "i.e. , at $t=0$ a parameter". i. e. would be advantageously replaced by e. g.

b) Beginning of page 5. " a part" should read "apart"

c) Page 30. A word is missing at the end of the sentence "but it would be very interesting to do."

  • validity: high
  • significance: high
  • originality: good
  • clarity: good
  • formatting: excellent
  • grammar: good

Author:  Vincenzo Alba  on 2018-02-24  [id 216]

(in reply to Report 1 on 2018-02-14)
Category:
answer to question

We would like to thank the referee for her/his work and for recommending the publication of our paper.
We also thank the referee for her/his suggestions that we implemented in the new version of the manuscript as
described below.

a) We now added above (4) the sentence

"In a generic interacting integrable model there are several families of quasiparticles. The generalization of (3)
is obtained by summing all the contributions of the different families."

Following the second referee suggestion we now added below (3)

"Notice that (3) does not take into account ${\mathcal O}(1)$ terms, which are subleading in the scaling limit."

b) We now moved the sentence in question below formula (2).

c) We implemented the referee's suggestion.

d) We would like to thank the referee for this remark. The presentation in section 3.1 is quite general and it can
be adapted to any integrable model with minor modifications. However, it is true that we have in mind the
spin-1/2 XXZ chain. This is now explicitly stated in the beginning of 3.1 as

"The prototype integrable model that we consider here is the XXZ spin-$1/2$ chain in the regime
with $\Delta>1$, although the TBA results that we will discuss can be generalized to other integrable
models with minor modifications."

To comment on the gapless case below (31) we added the sentence

"For the XXZ chain with \Delta < 1 the structure of the
string solutions is more complicated than (31), although major simplifications occur for
\Delta=\Delta_k= -\cos(\pi/k) with k = 1,2,\dots (roots of unity)."

e) We thank the referee for this remark. We now removed the word quench action since it is not relevant to
understand the content of the section.

f) To comment on the numerical problems to simulate the Lieb-Liniger model we added at page 21 the sentence

"We should mention that, in contrast with the XXZ chain, here we cannot provide a
numerical check of our preditions. This is due to the fact that as of now for models in the continuum there
are no efficient numerical methods, such tDMRG."

Finally, we would like to thank the referee for spotting the typos that we corrected in the new version of
the manuscript.

---

## Round 1 · Referee Report · Anonymous (Referee 2) · 2018-2-18

Strengths

  1. Quantitative picture of entanglement dynamics in interacting systems
  2. Good agreement with numerical results
  3. Well-written paper, results easy to follow

Weaknesses

Extended version of Ref. [10], with only a few new results

Report

In this paper, the authors study the evolution of the entanglement entropy after a quantum quench in integrable systems using a quasi-particle picture. This paper is essentially an extended version of Ref 10, including technical details and some new results. The main idea is to combine the quasi-particle picture of entanglement propagation put forward by Calabrese and Cardy in [6] with more quantitative ingredients accessible using Bethe ansatz results (quasi-particle velocities and Yang-Yang entropy in the steady state) to arrive at a quantitative prediction for the dynamics of entanglement. Remarkably, this simple picture that entanglement is generated by EPR pairs of quasi-particles emitted at the same point in the initial state appears to be quantitatively accurate. The authors explore this idea in detail and compare to numerical results for non-interacting and interacting systems. In the case of non-interacting systems, the approach is especially simple and the results can be compared to existing ab-initio results, and to numerical results. For interacting systems such as the XXZ spin chain, the authors explain review how the steady-state thermodynamic entropy and the group velocities can be computed using Bethe ansatz techniques, and they compare the predictions of the quasi-particle approach to numerical matrix product state calculations. This part of the paper follows closely Ref 10, but it includes more technical details and some new interesting discussions including a breakdown of the quasi-particle contributions to entanglement, and results on the dynamics of the mutual information. The authors also study the entanglement dynamics in the Lieb-Liniger model using their approach: it will be interesting to see if their results can be compared to numerics or to other approaches in the future.

I think this is an interesting, well-written paper that is suitable for publication in SciPost. In my opinion, it is quite remarkable that the quasi-particle picture of Ref 6 is quantitatively correct. I recommend publication as is.

Requested changes

In the introduction, the authors write:
``A major breakthrough in this respect has been achieved in [10] where it has been shown that, at least for certain classes of quenches in integrable models, the function $s(\lambda)$ can be conjectured from the equivalence between the entanglement and the thermodynamic entropy in the stationary state.’’
While I find the idea of Ref 10 to use the picture of Ref 6 more quantitatively very nice, I find it odd to put the emphasis on the relation between the entanglement and the thermodynamic entropy in the stationary state, which seems like the most natural part of this result. For any thermalizing system (in a generalized sense for integrable systems, which thermalize to GGEs), one would expect the entanglement entropy to coincide with the thermodynamic entropy in the steady state. In generic systems, this follows from ETH or from the very idea of thermalization. While it’s remarkable that (4) can be used to capture the entanglement dynamics, I see no reason to doubt that at long times, the entanglement entropy should coincide with the steady state entropy. Do the authors have any reason to question this?

  • validity: high
  • significance: high
  • originality: good
  • clarity: high
  • formatting: excellent
  • grammar: good

Author:  Vincenzo Alba  on 2018-02-24  [id 217]

(in reply to Report 2 on 2018-02-18)

We would like to thank the referee for her/his work, for the positive evaluation of the manuscript, and for considering our work interesting and timely.
The only criticism of the referee is that we put too much emphasis on the relation between the entanglement and
the thermodynamic entropy in the stationary state. We completely agree with the referee that the relation between the entanglement
and the thermodynamic entropy is natural, especially in the integrability community. However, outside of this community this
natural result might be under appreciated. Given that the paper aims at being understandable for the broadest possible audience
we would prefer, if the referee agrees, to keep the introduction as it is.

---

## Round 1 · Referee Report · Anonymous (Referee 3) · 2018-2-27

Strengths

1- very well explained 2- quite exhaustive in the models and effects it checks 3- excellent numerical verifications

Weaknesses

1- not a lot new, as it is mainly a long version of another paper proposing the formula 2- slightly repetitive reading

Report

In this paper the authors study the evolution of entanglement entropy in homogeneous quantum quenches. The main point is to study and verify numerically formula (4) which was proposed in [10], which gives full precision, in the case of Bethe integrable systems, on the "general structure" formula (3) proposed in [6]. Formula (3) was based on physically counting particle pairs creating entanglement between the region and its outside, as they evolve after a quench. Formula (4) gives meaning to these pairs via the quasiparticles of the thermodynamic Bethe ansatz. In the present paper, formula (4) is fully and exhaustively analyzed in a variety of models, including free models (in the free boson it seems there does not exist derivations of it), in the XXZ chain and in the Lieb-Liniger model (both attractive and repulsive). Numerical verification is giving in free models and in the XXZ chain, which are very convincing. The physics underlying the actual behaviour of the entanglement entropy as predicted by formula (4) is also analyzed, with contributions from various types of quasiparticles highlighted.

I believe this is a very good paper. Of course, it does not present fundamental new ideas, but its main purpose is to provide full analysis of ideas proposed previously, which it does well. It is a little bit boring to read, but this is in a sense by construction as per its main purpose. It is not too complicated, with emphasis given on the intuitive explanations instead of the technical details (although all technical details seem to be appropriately presented).

I do not have any proposed changes (only typo I found: "a part" -> "apart" top of page 5).

Requested changes

no change requested

---

## Editorial Decision

published